# Loss of *Kmt2c* or *Kmt2d* drives brain metastasis via KDM6A-dependent upregulation of MMP3

Marco Seehawer [1,2,3], Zheqi Li [1,2,3], Jun Nishida [1,2,3], Pierre Foidart[1,2,3], Andrew H. Reiter [4], Ernesto Rojas-Jimenez[1,2,3], Marie-Anne Goyette [1,2,3], Pengze Yan[1,2,3], Shaunak Raval[4], Miguel Munoz Gomez[5], Paloma Cejas [5], Henry W. Long [5], Malvina Papanastasiou [4] & Kornelia Polyak [1,2,3] ✉

*KMT2C* and *KMT2D*, encoding histone H3 lysine 4 methyltransferases, are among the most commonly mutated genes in triple-negative breast cancer (TNBC). However, how these mutations may shape epigenomic and transcriptomic landscapes to promote tumorigenesis is largely unknown. Here we describe that deletion of *Kmt2c* or *Kmt2d* in non-metastatic murine models of TNBC drives metastasis, especially to the brain. Global chromatin profiling and chromatin immunoprecipitation followed by sequencing revealed altered H3K4me1, H3K27ac and H3K27me3 chromatin marks in knockout cells and demonstrated enhanced binding of the H3K27me3 lysine demethylase KDM6A, which significantly correlated with gene expression. We identified *Mmp3* as being commonly upregulated via epigenetic mechanisms in both knockout models. Consistent with these findings, samples from patients with *KMT2C*-mutant TNBC have higher *MMP3* levels. Downregulation or pharmacological inhibition of KDM6A diminished *Mmp3* upregulation induced by the loss of histone–lysine *N*-methyltransferase 2 (KMT2) and prevented brain metastasis similar to direct downregulation of *Mmp3*. Taken together, we identified the KDM6A–matrix metalloproteinase 3 axis as a key mediator of KMT2C/D loss-driven metastasis in TNBC.

Breast cancer is the most common cancer and is a leading cause of cancer-related deaths in women worldwide[1]. It is classified, based on the expression of oestrogen and progesterone receptors and human epidermal growth factor receptor 2 (HER2), into hormone receptor-positive, HER2-positive or triple-negative breast cancer (TNBC). TNBC has the highest risk of distant metastases and the worst outcome among the major subtypes[2]. Approximately 30% of patients with metastatic TNBC have brain metastases associated with the shortest overall survival[3].

Sequencing of primary and metastatic tumours identified *KMT2C* as being more commonly mutated in distant metastases compared with

primary breast tumours[4]. Both *KMT2C* and *KMT2D* are also frequently mutated in breast cancer brain metastases[5], implying functional relevance. Histone–lysine *N*-methyltransferase 2C (KMT2C) and KMT2D function as histone H3 lysine 4 (H3K4) methyltransferases, converting unmethylated H3K4 to methylated H3K4 (H3K4me1) via monomethylation of lysine 4 on the histone H3 protein subunit. H3K4me1 is enriched in active or primed enhancers and promoters and enables recruitment of histone H3K27 acetyltransferases (for example, P300) and H3K27 demethylases (for example, lysine-specific demethylase 6A (KDM6A))[6]. KDM6A and KMT2C or KMT2D are components of

[1]Department of Medical Oncology, Dana-Farber Cancer Institute, Boston, MA, USA. [2]Department of Medicine, Brigham and Women's Hospital, Boston, MA, USA. [3]Department of Medicine, Harvard Medical School, Boston, MA, USA. [4]The Eli and Edythe L. Broad Institute, Cambridge, MA, USA. [5]Center for Functional Cancer Epigenetics, Dana-Farber Cancer Institute, Boston, MA, USA. ✉e-mail: kornelia_polyak@dfci.harvard.edu

the epigenetic regulatory complex of proteins associated with SET1 (COMPASS)[6], and KDM6A depends on KMT2C regardless of its catalytic activity[7,8]. Similarly, binding of KMT2D to other components of the COMPASS complex is independent of its enzymatic domain[9]. Thus, KMT2C and KMT2D are considered core proteins of the COMPASS complex, and deficiency in either of them affects epigenomic landscapes. Loss of *KMT2C* or *KMT2D* induces hybrid epithelial–mesenchymal transition (EMT) states and alters EMT balance, which can promote metastasis[10,11]. *KMT2C* deficiency in lung cancer represses DNA methyltransferase 3A expression, leading to DNA hypomethylation and a subsequent increase in metastasis[12]. However, it is unknown whether loss of *KMT2C* and *KMT2D* induces similar or unique epigenetic alterations and which underlying pathways might drive metastasis in *KMT2C*- or *KMT2D*-deficient tumours.

## Results

### Deficiency of *Kmt2c* or *Kmt2d* enables metastasis

*KMT2C* and *KMT2D* are among the top frequently mutated genes in TNBC, with mutations detected in 8.0 and 8.7% of tumours, respectively (Extended Data Fig. 1a). There are no mutational hotspots in the SET domain mediating enzymatic activity (Extended Data Fig. 1b), but most mutations are truncating or missense, implying loss of function. The expression of *KMT2C* and *KMT2D* is significantly reduced in distant metastases compared with matched primary TNBC (Fig. 1a), suggesting a role for the loss of KMT2C/D function in metastasis. Data from the DepMap database[13] also showed a significant inverse correlation between metastatic potential and *KMT2C* expression in breast cancer cell lines and a similar trend for *KMT2D* (Extended Data Fig. 1c).

To investigate the consequences of *Kmt2c* and *Kmt2d* deficiency, we deleted these genes in the non-metastatic 168FARN and 67NR mouse mammary tumour cell lines by transfecting an mCherry-tagged single guide RNA (sgRNA) pool and a plasmid expressing *Cas9* and green fluorescent protein (GFP) (Fig. 1b). Sequencing of targeted genomic regions showed 90% (*Kmt2c*) and 100% (*Kmt2d*) mutant reads in 168FARN cells (Extended Data Fig. 1d) and 64% (*Kmt2c*) and 98% (*Kmt2d*) mutant reads in 67NR cells (Extended Data Fig. 1e). Immunoblot analysis confirmed reduced protein expression in knockout (KO) derivatives of both cell lines (Fig. 1c and Extended Data Fig. 1f).

Characterization of the cells showed no differences in cell proliferation (Extended Data Fig. 1g) and a slight but significant decrease in the half-maximal inhibitory concentration for doxorubicin in 168FARN *Kmt2c* KO cells and for the KDM6A inhibitor GSK-J4 in 168FARN *Kmt2c* and *Kmt2d* KO cells (Extended Data Fig. 1h,i). We then transduced 168FARN cells with H2B–mCherry and injected them into the mammary fat pads (MFPs) of syngeneic immunocompetent female BALB/c mice (Fig. 1d). Loss of *Kmt2c* or *Kmt2d* had no significant effect on primary tumour growth (Fig. 1e), but we detected mCherry⁺ micro-metastatic lesions in the liver, lungs and brain of mice injected with KO but not wild-type (WT) cells (Fig. 1f,g). Although we established cultures from all primary tumours, mCherry⁺ cancer cells only grew in culture from distant organs of mice injected with *Kmt2c* and *Kmt2d* KO cells, consistent with our staining data (Fig. 1g). Immunoblot analyses of these cell cultures showed generally lower expression of KMT2C and KMT2D in metastases compared with matched primary tumours, suggesting that they were derived from KO cells. Importantly, there was no compensatory upregulation of KMT2C in *Kmt2d* KO cells and vice versa (Extended Data Fig. 1j,k).

To assess metastases in a more quantitative manner, we conducted intracardiac injections and performed flow cytometry from dissociated brain, liver, and lungs. mCherry⁺ cells were significantly enriched in the brains of mice injected with *Kmt2c* and *Kmt2d* KO but not WT derivatives of both 168FARN and 67NR lines, but not in liver or lung tissues (Fig. 1h,i and Extended Data Fig. 1l–n). To validate our detection method, we also used quantitative PCR to measure *mCherry* expression and found a significant correlation between the two assays (Extended Data Fig. 1o).

Haematoxylin and eosin staining confirmed macro-metastases in brains, consistent with Fig. 1h,i (Fig. 1j). Additionally, femoral bone sections revealed macro-metastases exclusively in *Kmt2c* and *Kmt2d* KO groups (Extended Data Fig. 1p,q), demonstrating the gain of a multi-organ metastatic phenotype following *Kmt2c* or *Kmt2d* loss.

### *Kmt2c* and *Kmt2d* KO-specific tumour immune microenvironment

Loss of *Kmt2d* sensitizes tumours to immune checkpoint inhibitors (ICIs)[14]. However, it is not known whether loss of *Kmt2c* and *Kmt2d* has a similar impact on the immune microenvironment and if this is responsible for the observed metastatic phenotype. Thus, we performed single-cell RNA sequencing (scRNA-seq) of primary tumours derived from MFP injection of 168FARN derivatives. Tumour epithelial cells were identified based on the expression of *mCherry* and the epithelial marker *Krt8*, whereas stromal cells were annotated using known cell type-specific markers (Fig. 2a, Extended Data Fig. 2a,b and Supplementary Table 1). Tumour cells showed clear separation of *Kmt2c* and *Kmt2d* KO from WT (Fig. 2b), and differential gene expression (DGE) analysis identified *Ly6a*, *Bst2*, *Ifi27l2a* and *Stat1* as the top highly upregulated genes in both KO tumour cells compared with the WT (Extended Data Fig. 2c,d and Supplementary Table 2). This implied a pro-inflammatory milieu via potential activation of interferon signalling, as confirmed by pre-ranked gene set enrichment analysis (Fig. 2c). However, several immune checkpoint genes, including *Cd274* (encoding programmed death ligand 1) were increased in KO compared with WT cells (Fig. 2d,e, Extended Data Fig. 2e and Supplementary Table 2). Upregulation of programmed death ligand 1 was also confirmed by immunofluorescence in both *Kmt2c* KO and *Kmt2d* KO primary tumours (Fig. 2f,g).

We then subclustered and annotated non-epithelial cells and grouped them according to genotype (Fig. 2h and Extended Data Fig. 2f,g). Fibroblasts were classified into inflammatory and myofibroblastic cancer-associated fibroblasts, and myofibroblastic cancer-associated fibroblast signatures were higher than inflammatory cancer-associated fibroblast ones in the *Kmt2c* KO but not the *Kmt2d* KO compared with the WT (Extended Data Fig. 2h,i). Endothelial cells from *Kmt2c* KO and *Kmt2d* KO tumours were distinct from the WT and shared top upregulated genes, including *Cd74*, *Cxcl9*, *H2-Ab1* and *H2-Eb2* (Fig. 2j,k). The most abundant immune cells were macrophages, with a higher ratio of M1/M2 macrophages in *Kmt2c* but not *Kmt2d* KO tumours compared with the WT (Extended Data Fig. 2l,m). T cell populations were classified into total CD4⁺, T regulatory, total CD8⁺, cytotoxic CD8⁺ and exhausted CD8⁺ (Extended Data Fig. 2n). The signature of immune-suppressive T regulatory cells was higher in *Kmt2d* KO tumours and showed an increasing trend in *Kmt2c* KO tumours compared with the WT (Fig. 2i–k). The ratio of total CD8⁺ T cells was higher in the *Kmt2c* KO; however, these cells had a trend of higher exhaustion than cytotoxic signatures (Fig. 2l–n). Immunofluorescence confirmed significantly more CD8⁺ T cells specifically in *Kmt2c* KO tumours compared with the WT (Fig. 2o,p). In line with this—CCL5, a chemoattractant for monocytes and T cells[15]—was only increased in *Kmt2c* but not *Kmt2d* KO cell supernatants and tumours, whereas CXCL1 and matrix metalloproteinase 3 (MMP3) were higher in all KO samples (Extended Data Fig. 2o).

We then performed intracardiac injection in immunodeficient NSG mice. Again, we detected mCherry⁺ cancer cells only in the brains of mice injected with *Kmt2c* or *Kmt2d* KO cells but not WT cells (Fig. 2q). This suggested that *Kmt2c* or *Kmt2d* loss-associated immune suppression might not be the main driver of their metastatic phenotype.

### *Kmt2c* and *Kmt2d* KO changes chromatin and the transcriptome

To dissect *Kmt2c* or *Kmt2d* loss-induced cell intrinsic changes, we first focused on histones as the direct targets of KMT2C and KMT2D.

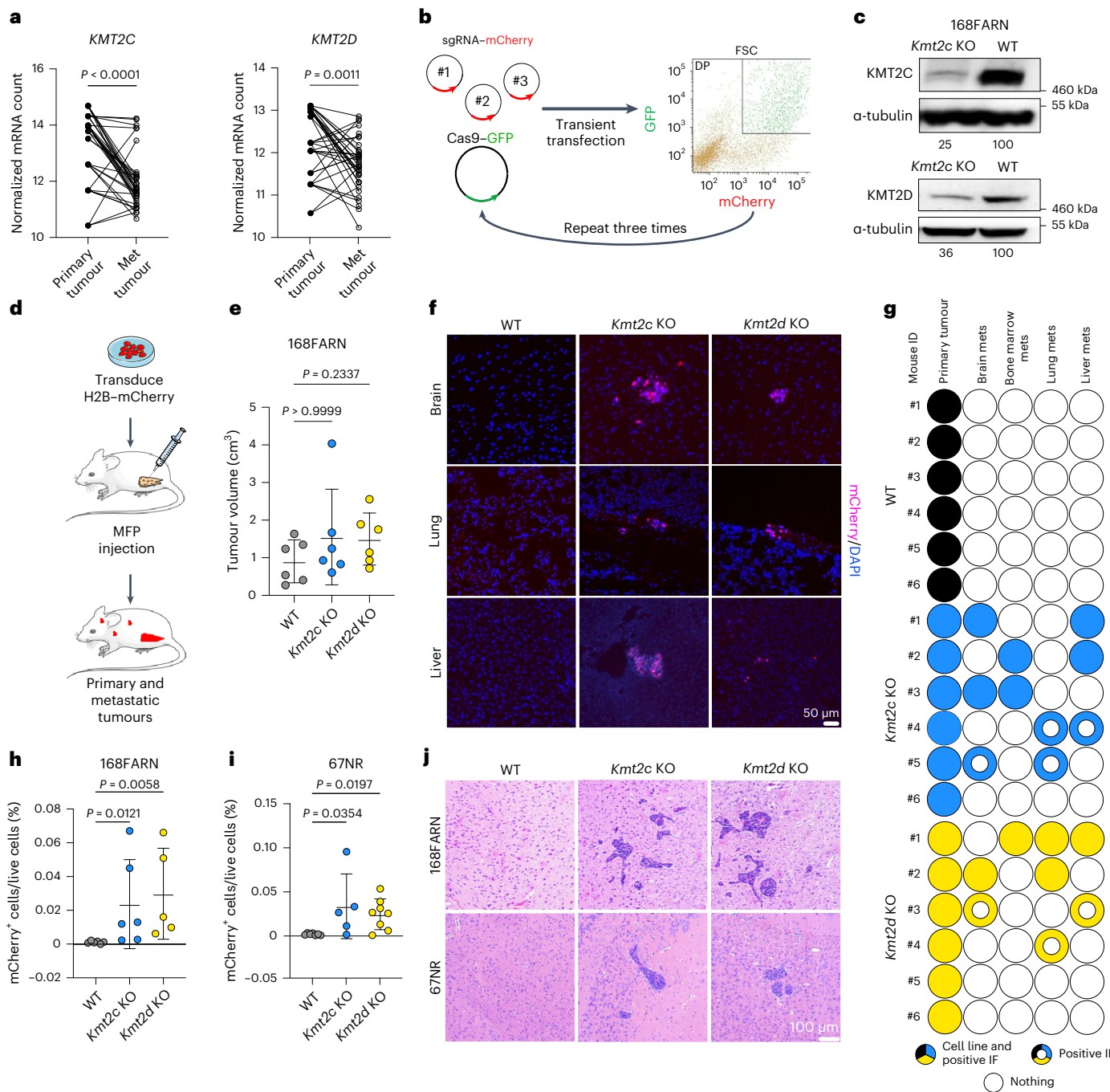

**Fig. 1 | Loss of *Kmt2c* or *Kmt2d* induces multi-organ metastasis. a**, Normalized messenger RNA (mRNA) counts for *KMT2C* and *KMT2D* in matched primary and metastatic (Met) TNBC/basal-like tumours. Data taken from ref. 30 (*n* = 37 pairs from nine patients). Statistical significance was determined by two-tailed paired *t*-test (*P* = 2.6 × 10$^{-6}$ for *KMT2C* KO). **b**, Schematic of the generation of KO cell lines using transient transfection and fluorescence-activated cell sorting. **c**, Immunoblot analyses of KMT2C and KMT2D in WT and KO cell lines (*n* = 2 biological replicates, numbers indicate arbitrary units of densitometric measurements of indicated blots normalized to tubulin and WT). **d**, Schematic of the spontaneous metastasis model. **e**, Graph depicting primary tumour volumes 19 d after MFP injection of KO and WT cells into BALB/c mice (*n* = 6 per group). Significance was determined by Kruskal–Wallis test with Dunn's multiple comparison and the results are presented as means ± s.d. **f**, Immunofluorescence for mCherry (magenta) and 4′,6-diamidino-2-phenylindol (DAPI; blue) in the brain, liver and lungs from the mice used in **e**. Scale bar, 50 μm. **g**, Schematic

of samples with mCherry⁺ cells (from two individual sections per sample) and cell lines derived from primary tumours and organ-specific metastases (mets) from the mice depicted in **e**. The colours indicate WT (black), *Kmt2c* KO (blue) and *Kmt2d* KO (yellow) cells. The doughnuts represent samples with mCherry⁺ staining but no cell line established. **h,i**, Plots showing the quantification of mCherry⁺ cells from the brain tissue of mice 12 d after intracardiac injection with 168FARN KO or WT cells (**h**; *n* = 6 (WT and *Kmt2c* KO) or 5 (*Kmt2d* KO) mice per group) or 13 d after injection with 67NR KO or WT cells (**i**; *n* = 7 (WT), 5 (*Kmt2c* KO) or 8 (*Kmt2d* KO) mice per group). Statistical significance was determined by Kruskal–Wallis test with Dunn's multiple comparison and the data are presented as means ± s.d. **j**, Representative haematoxylin and eosin-stained brain sections of the mice depicted in **h** and **i**. Scale bar, 100 μm. DP, double positive; FSC, forward scatter; IF, immunofluorescence. Panel **d** created with images from Servier Medical Art under a Creative Commons license CC BY 4.0.

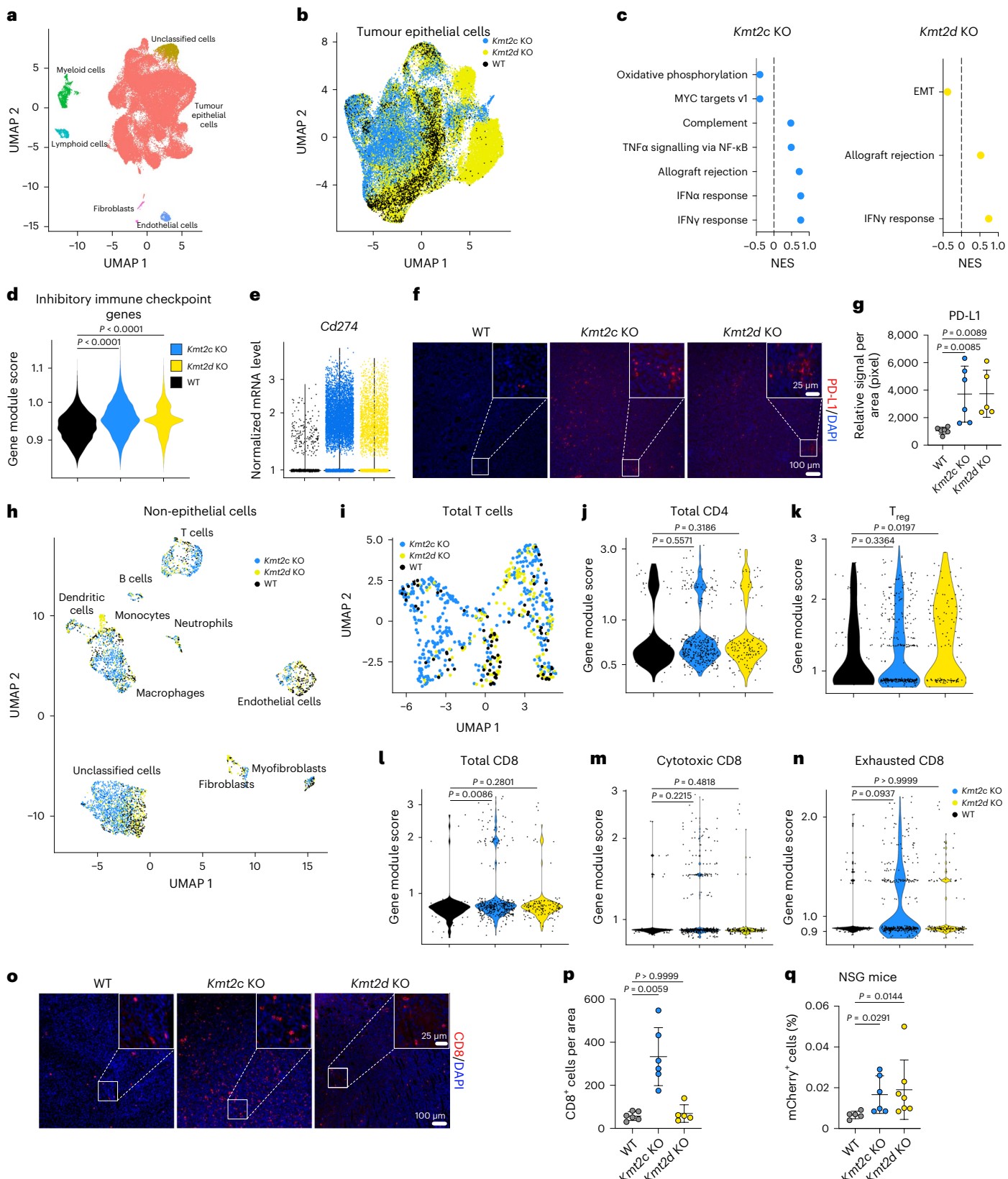

Quantitative histone mass spectrometry showed similar histone modifications in both KO cells compared with the WT (Fig. 3a). H3K4me1 and trimethylation of lysine 27 on the histone H3 protein subunit (H3K27me3) were significantly decreased in the *Kmt2c* KO but not in the *Kmt2d* KO, whereas acylation of lysine 27 on the histone H3 protein subunit (H3K27ac) was significantly increased in both KO lines (Fig. 3b).

Immunoblot analysis showed similar levels of H3K4me1, but decreased H3K27me3 and increased H3K27ac in both KO lines (Fig. 3c).

To explore alterations in chromatin profiles, we performed chromatin immunoprecipitation followed by sequencing (ChIP-seq) for H3K4me1, H3K27ac, and H3K27me3. We found significantly gained (7,367 for *Kmt2c* KO and 9,824 for *Kmt2d* KO) or lost (4,085 for *Kmt2c* KO

**Fig. 2 | Distinct impact of *Kmt2c* and *Kmt2d* loss on the tumour immune microenvironment. a**, Uniform manifold approximation and projection (UMAP) of scRNA-seq of all cells from primary tumours of *Kmt2c* KO, *Kmt2d* KO and WT cells. Clusters are coloured and annotated based on gene module score. **b**, UMAP of subclustered tumour cells coloured by genotype. **c**, Normalized enrichment scores (NESs) of fast gene set enrichment analyses using differential genes in *Kmt2c* KO (left) or *Kmt2d* KO (right) tumour cells compared with the WT. Only hits with an adjusted *P* value of <0.1 are shown. **d**, Violin plot of gene module scores for immune checkpoint gene expression in WT, *Kmt2c* KO or *Kmt2d* KO cancer cells. Statistical significance was determined by Wilcoxon test with Benjamini–Hochberg multiple comparison ($P = 2.2 \times 10^{-16}$). **e**, Expression levels of *Cd274* in cancer cells from scRNA-seq in WT, *Kmt2c* KO and *Kmt2d* KO cells. **f**, Representative images of immunofluorescence staining of programmed death ligand 1 (PD-L1; red) and DAPI (blue) in the tumours depicted in Fig. 1e. Main scale bars, 100 μm. Scale bars in insets, 25 μm. **g**, Quantification of the average PD-L1⁺ area per field of view from three of four independent images per tumour (*n* = 6 (WT and *Kmt2c* KO) or 5 (*Kmt2d* KO) mice per group). Statistical significance was determined by Kruskal–Wallis test with Dunn's multiple comparison and

the data are presented as means ± s.d. **h**, UMAP of non-tumour cells annotated using gene module scores and coloured by genotype. **i**, UMAP of subclustered total T cells coloured by genotype. **j**–**n**, Gene module scores for total CD4 (**j**), T regulatory cell (T_reg; **k**), total CD8 (**l**), cytotoxic CD8 (**m**) and exhausted CD8 (**n**) for WT, *Kmt2c* KO and *Kmt2d* KO samples. Statistical significance was determined by Kruskal–Wallis test with Dunn's multiple comparison. **o**, Representative images of immunofluorescence staining of CD8 (red) and DAPI (blue) in the tumours depicted in Fig. 1e. Main scale bars, 100 μm. Scale bars in insets, 25 μm. **p**, Quantification of average CD8⁺ cells per field of view from three or four independent images per tumour (*n* = 6 (WT and *Kmt2c* KO) or 5 (*Kmt2d* KO) mice per group). Statistical significance was determined by Kruskal–Wallis test with Dunn's multiple comparison and the data are presented as means ± s.d. **q**, Quantification of mCherry⁺ cells in the brain suspension of NSG mice 11 d after cardiac injection with *Kmt2c* KO, *Kmt2d* KO or WT cells (*n* = 5 (WT), 6 (*Kmt2c* KO) or 7 (*Kmt2d* KO) mice per group). Statistical significance was determined by Kruskal–Wallis test with Dunn's multiple comparison and the data are presented as means ± s.d. IFN, interferon; NF-κB, nuclear factor κB; TNF, tumour necrosis factor.

and 4,896 for *Kmt2d* KO) H3K4me1 peaks (Fig. 3d and Supplementary Table 3), implying locus-specific rather than global changes in H3K4me1. Intersection of gained or lost peaks between *Kmt2c* and *Kmt2d* KO showed more unique than shared peaks (Fig. 3e). H3K27ac ChIP-seq also showed significantly gained (5,656 for *Kmt2c* KO and 6,031 for *Kmt2d* KO) and lost (3,992 for *Kmt2c* KO and 3,866 for *Kmt2d* KO) peaks in both KO cell lines with again more unique than shared changes (Fig. 3f,g and Supplementary Table 3). Because KMT2C and KMT2D are H3K4 methyltransferases, changes in H3K27ac might be an indirect effect of their KO. Thus, we performed ChIP-seq for the H3K27 acetyl transferase P300 but only detected a few significantly gained peaks (401 for *Kmt2c* KO and 36 for *Kmt2d* KO). However, overall changes of P300 and H3K27ac signal intensities were significantly correlated in KO cells (Extended Data Fig. 3b). Moreover, the P300 signal intensity was significantly higher in gained H3K27ac peaks compared with unchanged peaks in both *Kmt2c* and *Kmt2d* KO cells and lower in lost H3K27ac peaks in *Kmt2c* KO cells (Fig. 3h). An increase in H3K27ac can be a consequence of a decrease in H3K27 methylation, thus we performed H3K27me3 ChIP-seq. There were no significantly changed peaks, probably due to the overall lower number of total called peaks in the *Kmt2c* and *Kmt2d* KO lines compared with the WT (Fig. 3i and Extended Data Fig. 3c). However, consistent with our immunoblot analyses (Fig. 3a–c), the global H3K27me3 signal was decreased in both KO lines (Fig. 3i).

The H3K27me3 demethylase KDM6A is part of the KMT2C/D COMPASS complex. Thus, we performed KDM6A ChIP-seq, which identified significantly gained (39,467 for *Kmt2c* KO and 5,234 for

*Kmt2d* KO) and very few lost (341 for *Kmt2c* KO and 1,003 for *Kmt2d* KO) peaks in KO cells compared with WT ones (Fig. 3j and Supplementary Table 3). Intersection again showed more unique than shared gained KDM6A peaks between the two KO cell lines (Fig. 3k). Motif analysis for each ChIP-seq dataset showed shared motifs, including ZNF263 and HOXA1 for gained H3K4me1 and H3K27ac, respectively. The Tlx and HIC1 motif was enriched in gained KDM6A peaks, suggesting a putative role of these factors upon loss of *Kmt2c* or *Kmt2d* (Extended Data Fig. 3d). Intersection of gained H3K27ac and gained KDM6A peaks within *Kmt2c* and *Kmt2d* KO cells showed minimal overlap (Extended Data Fig. 3e), highlighting P300 as a mediator of the observed differences in H3K27ac.

We then performed RNA-seq of *Kmt2c* and *Kmt2d* KO cells and identified DGEs compared with the WT (Fig. 4a and Supplementary Table 4). The topmost significantly upregulated genes in 168FARN *Kmt2c* KO cells included *Mmp3*, *Scara5* and *Cerk*, whereas *Cfh*, *Pdzrn3* and *Scara5* were most significantly upregulated in *Kmt2d* KO cells. Of note, *Kdm6a* expression was similar in all genotypes. Overall, DGEs were more unique than shared between both KO lines (Fig. 4b, Extended Data Fig. 4a,b and Supplementary Table 4). Consistently, gene set enrichment analysis for Hallmark pathways showed apical surface, apoptosis and coagulation enriched in the *Kmt2c* KO and coagulation and angiogenesis in the *Kmt2d* KO (Extended Data Fig. 4c). However, prediction for transcription regulators of upregulated genes using the LISA algorithm[16] showed NR3C1, MED1 and SMARCA4 among the top for both KO lines (Extended Data Fig. 4d). Interestingly, in 67NR cells, significantly changed genes and enriched pathways were more

**Fig. 3 | Altered histone and KDM6A occupancy in *Kmt2c* and *Kmt2d* KO cells. a**, Heatmap of global chromatin profiling for WT, *Kmt2c* KO and *Kmt2d* KO cells. The values are log₂ transformed and normalized to the WT (*n* = 3 biological replicates). **b**, Peak area values for H3K4me1, H3K27me3 and H3K27ac from **a** (*n* = 3 biological replicates). Statistical significance was determined by one-way analysis of variance (ANOVA) with multiple comparisons for H3K4me1 (*n* = 4 histone marks per three biological replicates) and two-way ANOVA with multiple comparisons for H3K27me3 ($P = 1.51 \times 10^{-8}$ for *Kmt2c* KO versus WT) and H3K27ac ($P = 2 \times 10^{-10}$ for *Kmt2c* KO versus WT and $P = 4.3 \times 10^{-5}$ for *Kmt2d* KO versus WT). The results are presented as means ± s.d. **c**, Immunoblot for H3K4me1, H3K27me3 and H3K27ac from WT, *Kmt2c* KO and *Kmt2d* KO 168FARN cells (*n* = 2 biological replicates). **d**, Heatmap showing significantly changed H3K4me1 peaks in *Kmt2c* KO and *Kmt2d* KO compared with WT cells centred around the peak centre ± 3 kilobases (kb). Summary plots for changed peaks (*n* = 3 biological replicates) are shown at the top. **e**, Intersections of significantly changed H3K4me1 peaks between *Kmt2c* KO and *Kmt2d* KO cells compared with WT cells. **f**, Heatmap showing significantly changed H3K27ac peaks in *Kmt2c* KO or *Kmt2d* KO compared with WT cells centred around the peak centre ± 3 kb.

Summary plots for changed peaks (*n* = 3 biological replicates) are shown at the top. **g**, Intersections of significantly changed H3K27ac peaks between *Kmt2c* KO and *Kmt2d* KO cells compared with WT cells. **h**, Box plot depicting the average P300 signal intensity (bin counts) in lost, unchanged or gained H3K27ac peaks in *Kmt2c* KO and *Kmt2d* KO cells. Statistical significance was determined by Kruskal–Wallis test with Dunn's multiple comparison for each comparison (*n* = 3 biological replicates for H3K27ac; *n* = 2 biological replicates for P300; $P = 3.3 \times 10^{-8}$ for lost versus unchanged for *Kmt2c* versus WT and $P = 2.2 \times 10^{-16}$ for lost versus gained and lost versus unchanged for *Kmt2c* KO and *Kmt2d* KO versus WT). **i**, Heatmaps of H3K27me3 ChIP-seq in WT, *Kmt2c* KO and *Kmt2d* KO cells from all of the called peaks scaled from the transcription start site (TSS) to the transcription end site (TES) with an interval of ±10 kb (*n* = 2 biological replicates). **j**, Heatmap showing significantly changed KDM6A ChIP-seq in *Kmt2c* KO or *Kmt2d* KO compared with WT cells centred around the peak centre ± 3 kb. Summary plots for changed peaks (*n* = 2 biological replicates) are shown at the top. **k**, Intersections of significantly gained KDM6A peaks between *Kmt2c* KO and *Kmt2d* KO cells compared with WT cells.

similar between *Kmt2c* and *Kmt2d* KO cells (Extended Data Fig. 4e–g and Supplementary Table 4). EMT was the top enriched pathway in *Kmt2c* and *Kmt2d* KO 67NR cells, consistent with previous studies[11]. Of note, EMT was not detected in 168FARN KO cells (Extended Data Fig. 4c), probably because the parental cells were already mesenchymal. However, we found significant likelihoods of common deregulation when we overlapped differential genes between 67NR and 168FARN cell lines, suggesting that loss of *Kmt2c* or *Kmt2d* induces similar expression changes independent of the cell line (Extended Data Fig. 4h).

To assess the impact of altered chromatin on the transcriptome, we integrated RNA-seq and ChIP-seq data using binding and expression target analysis (BETA)[17]. Gained H3K4me1, H3K27ac and KDM6A peaks significantly correlated with upregulated genes, whereas lost H3K4me1 and H3K27ac peaks significantly correlated with downregulated genes in *Kmt2c* KO and *Kmt2d* KO cells, respectively (Fig. 4c). Lastly, we overlapped genes with correlations of upregulation from BETAs in *Kmt2c* or *Kmt2d* KO cells. The largest gene set was shared between all three BETAs, suggesting similar target gene regulation by H3K4me1, H3K27ac and KDM6A (Fig. 4d).

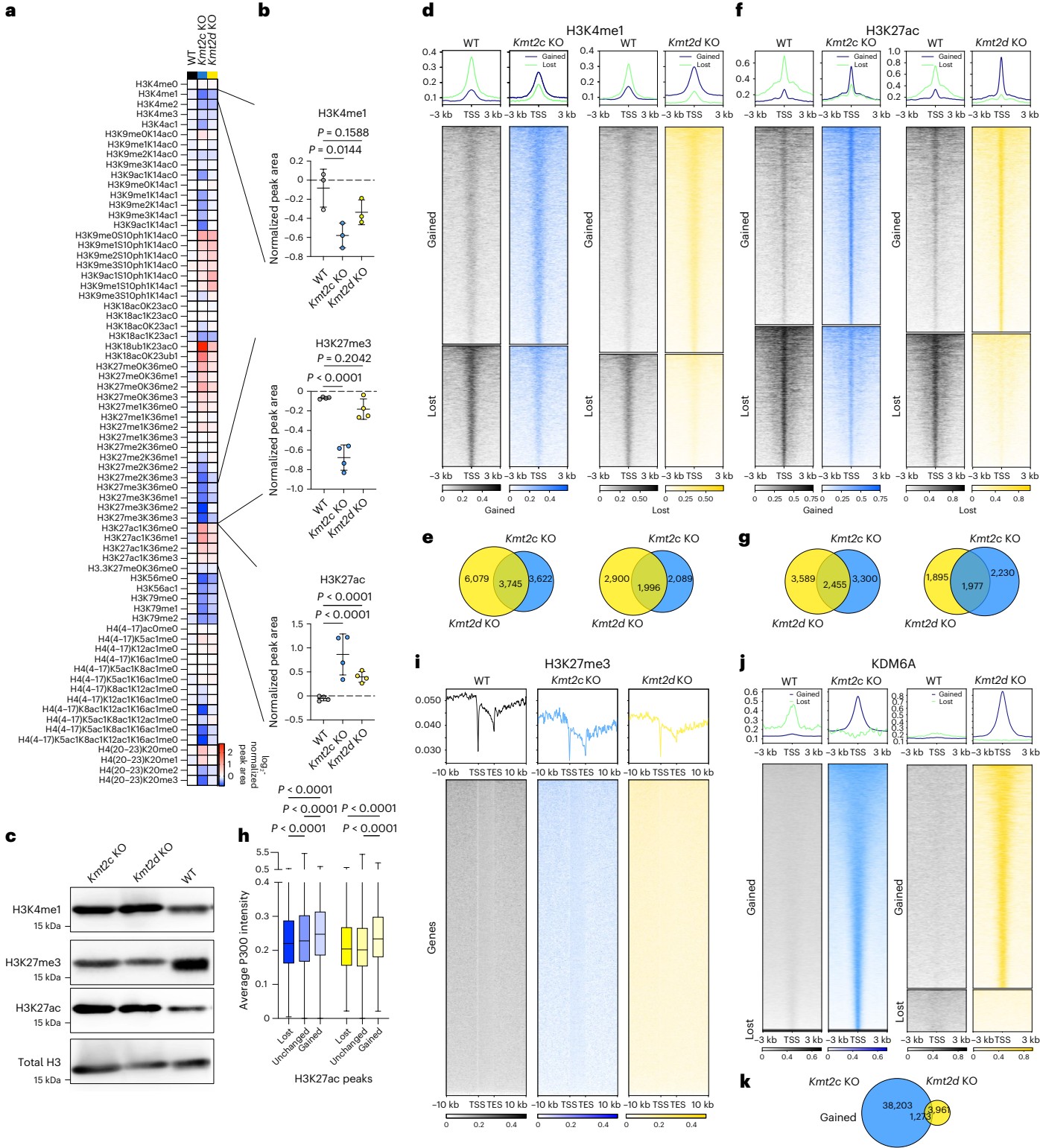

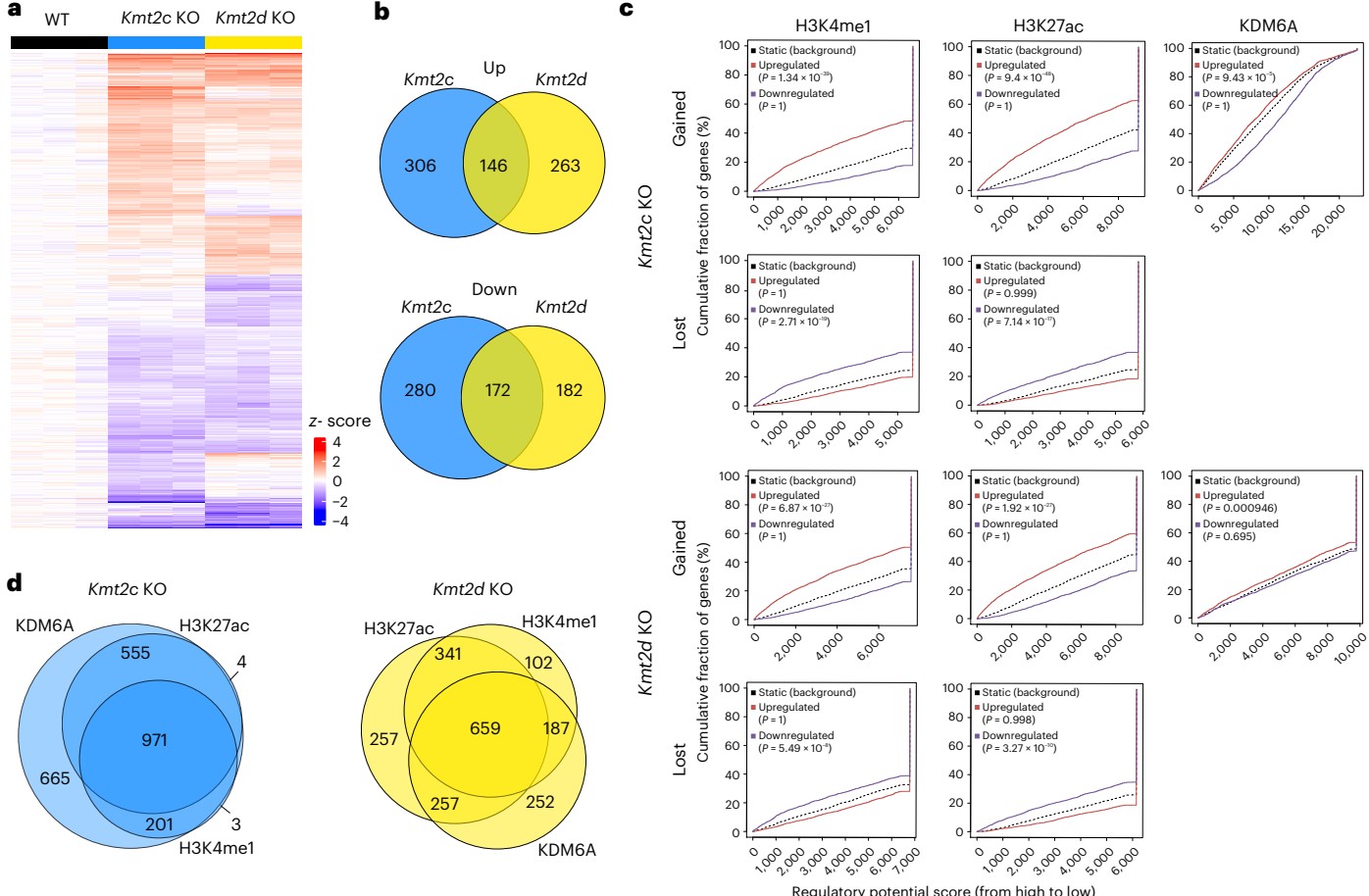

**Fig. 4 | KDM6A and histone remodelling correlates with transcriptomic changes upon *Kmt2c* and *Kmt2d* KO. a**, Heatmap depicting the expression of significantly changed genes in RNA-seq data of WT, *Kmt2c* KO and *Kmt2d* KO cells. Shown are batch-corrected *z* scores normalized to average WT counts (*n* = 3 biological samples per group). **b**, Venn diagrams showing the overlap of significantly up- or downregulated genes between *Kmt2c* KO (blue) and *Kmt2d* KO (yellow) compared with WT cells. **c**, Correlation plots from BETAs

for gained H3K4me1 ($P = 1.34 \times 10^{-39}$ and $6.87 \times 10^{-27}$), H3K27ac ($P = 9.4 \times 10^{-48}$ and $1.92 \times 10^{-27}$) and KDM6A ($P = 9.43 \times 10^{-5}$ and 0.000946) and lost H3K4me1 ($P = 2.71 \times 10^{-19}$ and $5.49 \times 10^{-8}$) and H3K27ac ($P = 7.14 \times 10^{-17}$ and $P = 3.27 \times 10^{-10}$) from *Kmt2c* KO and *Kmt2d* KO cells, respectively. **d**, Overlap of upregulated genes associated with gained H3K4me1, H3K27ac or KDM6A peaks extracted from **c** for *Kmt2c* KO or *Kmt2d* KO cells.

## Blocking KDM6A decreases brain metastasis by decreasing *Mmp3*

To identify common epigenetically regulated drivers of metastatic phenotype of both *Kmt2c* and *Kmt2d* KO cells, we integrated significantly gained H3K4me1 and H3K27ac peaks and genes significantly upregulated in RNA-seq and scRNA-seq of cancer cells, identifying *Mmp3* as the only overlapping gene (Fig. 5a). We confirmed enriched H3K4me1 and H3K27ac signal intensities at the *Mmp3* locus (Fig. 5b) and significantly upregulated *Mmp3* expression in both 168FARN (Fig. 5c) and 67NR KO derivatives (Fig. 5d). Of note, MMP3 was also a top hit in our secreted protein analysis (Extended Data Fig. 2o). Importantly, *MMP3* expression is significantly higher in human TNBC with *KMT2C* mutations compared with WT tumours, and *MMP3*-high tumours have significantly higher frequencies of *KMT2C* mutation (Fig. 5e and Extended Data Fig. 5a).

To functionally characterize *Mmp3*, we downregulated it using small hairpin RNA (shRNA) in 168FARN *Kmt2c* and *Kmt2d* KO and WT cells (Extended Data Fig. 5b). The cells were injected intracardially into mice and metastatic cells were quantified by flow cytometry. *Mmp3* downregulation significantly reduced brain metastases in both *Kmt2c* and *Kmt2d* KO cells (Fig. 5f,g). Previous clinical trials testing MMP inhibitors showed high toxicity and low efficacy[18]. However, our data suggest that KDM6A might be a mediator of *Kmt2c* and *Kmt2d* loss-associated

transcriptomic changes (Fig. 3c,d); thus, inhibiting KDM6A might be an indirect way of targeting MMP3. We therefore generated 168FARN *Kmt2c* and *Kmt2d* KO and WT cells expressing *Kdm6a*-targeting shRNAs (Extended Data Fig. 5c). *Kdm6a* downregulation led to a significant decrease in *Mmp3* expression (Fig. 6a) and significantly fewer brain metastases of *Kmt2c* KO and *Kmt2d* KO cells (Fig. 6b,c). The frequencies of bone metastases were not affected by knockdown of *Mmp3* or *Kdm6a* (Extended Data Fig. 5d). Similarly, downregulation of neither *Mmp3* nor *Kdm6a* had a significant effect on primary mammary tumour growth (Extended Data Fig. 5e).

To explore the potential mechanisms of *Mmp3* upregulation in *Kmt2c* and *Kmt2d* KO cells, we first treated cells with inhibitors of KDM6A (GSK-J4), KDM1A (ORY-1001) and P300 (A-485). GSK-J4 treatment decreased H3K27ac in *Kmt2c* and *Kmt2d* KO but not WT 168FARN cells without affecting H3K27me3 (Extended Data Fig. 5f). Similar to *Kdm6a* downregulation, GSK-J4 treatment significantly reduced *Mmp3* expression in the 168FARN and 67NR cell lines of both KOs, but not in WT 168FARN and 67NR cells (Fig. 6d and Extended Data Fig. 5g). Inhibition of KDM1A increased H3K4me1 in *Kmt2c* and *Kmt2d* KO cells but not in WT cells, while P300 inhibition decreased H3K27ac in all cell lines (Extended Data Fig. 6a). KDM1A inhibition did not affect *Mmp3* levels, whereas P300 inhibition significantly reduced *Mmp3* expression, specifically in *Kmt2c* and *Kmt2d* KO cells (Fig. 6e). Quantification

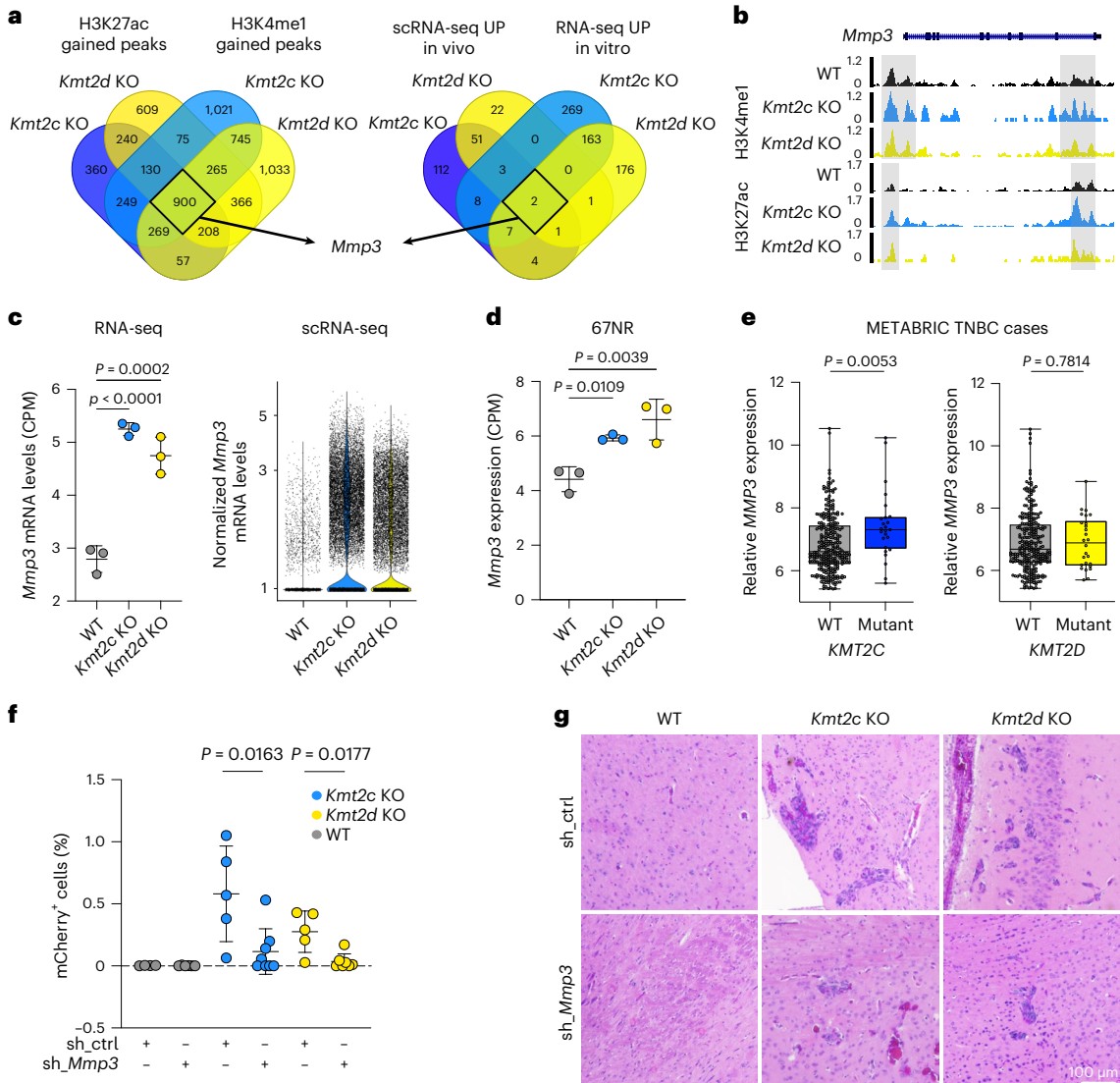

**Fig. 5 | *Mmp3* drives brain metastasis of *Kmt2c* and *Kmt2d* KO cells. a**, Overlap of genes with significant changes from Fig. 3d,f and Extended Data Figs. 2c,d and 4a,b. **b**, Genome tracks of H3K4me1 and H3K27ac peaks in WT, *Kmt2c* KO and *Kmt2d* KO cells at the *Mmp3* gene locus, with outlined peaks (H3K4me1 first peak: *P* = 0.0114 (*Kmt2c* KO) and *P* = 0.0005 (*Kmt2d* KO); H3K4me1 second peak: *P* < 0.0001 (*Kmt2c* KO) and *P* < 0.0001 (*Kmt2d* KO); H3K27ac first peak: *P* < 0.0001 (*Kmt2c* KO) and *P* = 0.0012 (*Kmt2d* KO); H3K27ac second peak: *P* < 0.0001 (*Kmt2c* and *Kmt2d* KO). Statistical significance was determined by Wald test. **c**, Counts per million (CPM) values for *Mmp3* from the RNA-seq data depicted in Extended Data Fig. 4a (left) and expression levels of *Mmp3* from the tumour cells in the scRNA-seq data depicted in Extended Data Fig. 2c,d (right) (*n* = 3 biological replicates). Statistical significance was determined by one-way ANOVA with multiple comparisons and the data are presented as means ± s.d. **d**, *Mmp3* mRNA expression in *Kmt2c* KO, *Kmt2d* KO and WT 67NR cells from the RNA-seq depicted in Extended Data Fig. 3f (*n* = 3 biological replicates). Statistical significance

was determined by one-way ANOVA with multiple comparison and the data are presented as means ± s.d. **e**, *MMP3* expression from the TNBC Molecular Taxonomy of Breast Cancer International Consortium (METABRIC) cohort with or without *KMT2C* or *KMT2D* mutation (*n* = 275 WT *KMT2C*; *n* = 25 mutated *KMT2C*; *n* = 274 WT *KMT2D*; *n* = 26 mutated *KMT2D*). The boxes represent the upper 75% percentile (top), median (line) and lower 25% percentile (bottom) and the whiskers range from minimum to maximum values. Statistical significance was determined by two-tailed Mann–Whitney *U*-test. **f**, mCherry⁺ cells in the brains from mice 11 d after intracardiac injection with control or sh_*Mmp3* derivatives of WT, *Kmt2c* KO or *Kmt2d* KO cells (*n* = 4 (WT sh_ctrl), 5 (*Kmt2c* KO sh_ctrl and *Kmt2d* KO sh_ctrl), 6 (WT sh_*Mmp3*), 7 (*Kmt2d* KO sh_*Mmp3*) or 8 (*Kmt2c* KO sh_*Mmp3*) mice per group). Statistical significance was determined by two-tailed Mann–Whitney *U*-test and the data are presented as means ± s.d. **g**, Haematoxylin and eosin-stained sections of the brains used in **f**. Scale bar, 100 μm. UP, upregulated.

---

of reads per million in our ChIP-seq data in the *Mmp3* promoter region showed a consistent decrease of H3K27me3 and an increase of P300 in *Kmt2c* and *Kmt2d* KO cells compared with WT cells, whereas KDM6A did not show consistent changes, indicating that KDM6A does not affect *Mmp3* expression via promoter binding (Extended Data Fig. 6b). The MMP gene cluster locus containing *MMP1*, *MMP3*, *MMP10*, *MMP12* and *MMP13* has been shown to be coordinately regulated, putatively by distal elements, resulting in correlative expression patterns[19]. Accordingly, we found a significant increase of *Mmp1b*, *Mmp10* and *Mmp13* in *Kmt2c* KO cells and a significant increase of *Mmp10* and

*Mmp13* in *Kmt2d* KO cells (Extended Data Fig. 6c). Fold changes of signal intensities within this cluster again showed increased H3K4me1, H3K27ac and P300 and decreased H3K27me3, whereas KDM6A did not show consistent changes (Extended Data Fig. 6d,e). However, inhibition of KDM6A or P300 decreased the expression of *Mmp1b* and *Mmp10* in *Kmt2c* and *Kmt2d* KO cells, supporting a role for distal regulatory functions of KDM6A (Extended Data Fig. 6f). In line with this, KDM6A binding exclusively in promoter regions does not correlate with upregulated gene expression (Extended Data Fig. 6g). However, there was a progressively increased significance of correlations

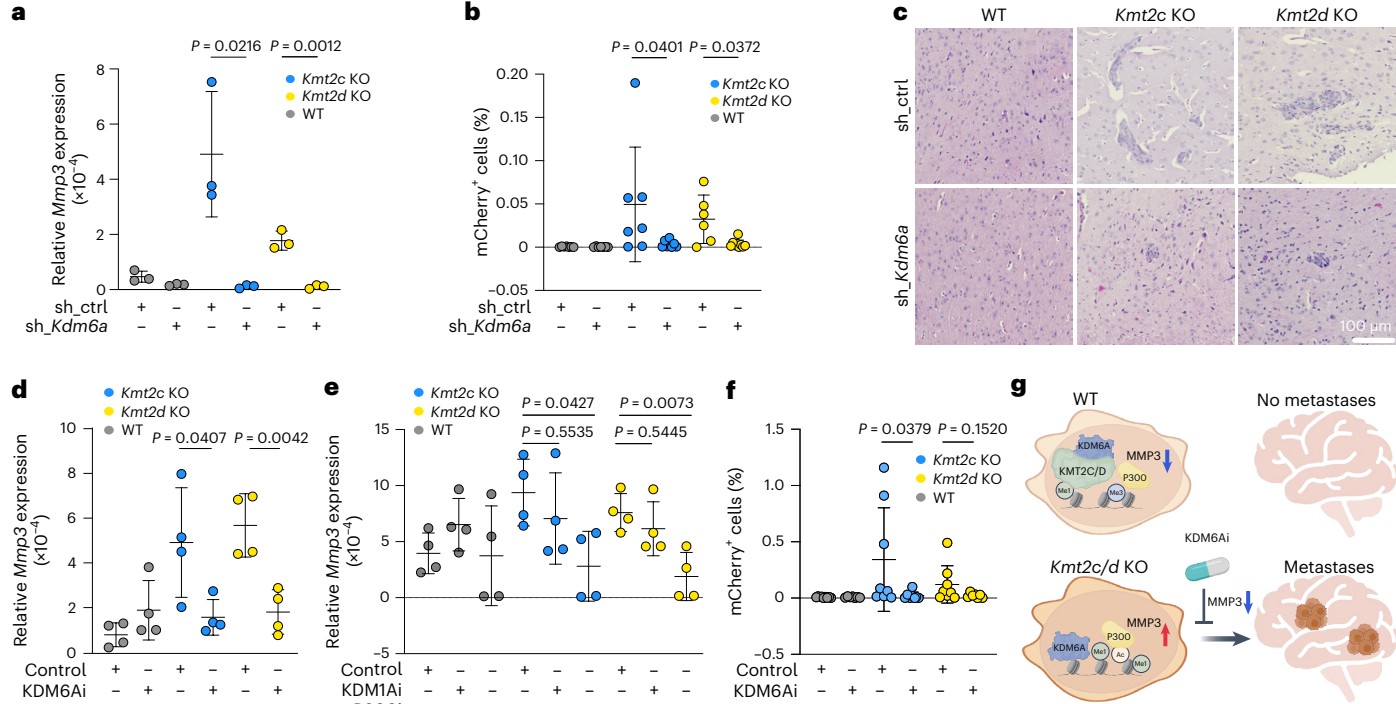

**Fig. 6 | Inhibition of KDM6A reduces *Mmp3* expression and brain metastases induced by *Kmt2c* or *Kmt2d* KO. a**, Relative expression of *Mmp3* in control and sh_*Kdm6a* knockdown derivatives of WT, *Kmt2c* KO and *Kmt2d* KO cells (*n* = 3 biological replicates). Statistical significance was determined by two-tailed unpaired *t*-test and the results are presented as means ± s.d. **b**, mCherry⁺ cells in brains from mice 11 d after intracardiac injection with control or sh_*Kdm6a* derivatives of WT, *Kmt2c* KO or *Kmt2d* KO cells (*n* = 6 (*Kmt2d* KO sh_ctrl), 7 (WT sh_ctrl, WT sh_*Kdm6a* and *Kmt2c* KO sh_*Kdm6a*), 8 (*Kmt2c* KO sh_*Kdm6a*) or 9 (*Kmt2d* KO sh_*Kdm6a*) mice per group). Statistical significance was determined by two-tailed Mann–Whitney *U*-test and the results are presented as means ± s.d. **c**, Haematoxylin and eosin-stained slides from the brains used in Fig. 6b. Scale bar, 100 μm. **d**, Relative expression of *Mmp3* in 168FARN WT, *Kmt2c* KO and *Kmt2d* KO cells treated for 3 d with dimethyl sulfoxide (DMSO; control) or GSK-J4 (a KDM6A inhibitor (KDM6Ai)) (*n* = 4 biological replicates). Statistical significance

was determined by two-tailed unpaired *t*-test and the results are presented as means ± s.d. **e**, Relative expression of *Mmp3* in 168FARN WT, *Kmt2c* KO and *Kmt2d* KO cells treated for 3 d with DMSO (control), ORY-1001 (KDM1Ai) or A-485 (P300i) (*n* = 4 biological replicates). Statistical significance was determined by one-way ANOVA with Dunnett's multiple comparison test and the results are presented as means ± s.d. **f**, mCherry⁺ cells in the brains of mice treated with either DMSO (control) or GSK-J4 (KDM6Ai) 12 d after intracardiac injection with DMSO or GSK-J4 pre-treated WT, *Kmt2c* KO or *Kmt2d* KO cells (*n* = 7 (WT GSK-J4 and *Kmt2d* KO GSK-J4), 8 (*Kmt2d* KO DMSO, *Kmt2c* KO DMSO and *Kmt2c* KO GSK-J4) or 9 (WT DMSO) mice per group). Statistical significance was determined by two-tailed Mann–Whitney *U*-test and the results are presented as means ± s.d. **g**, Schematic of the predicted mechanism of KMT2C/D mutation-induced histone remodeling and metastases induction through MMP3 and its prevention via KDM6A inhibition. Schematic in **g** created with BioRender.com.

when assessing KDM6A peaks with increasing distances from the transcription start site (Extended Data Fig. 6h). Mutations in *KMT2D* have been shown to disrupt COMPASS complex formation[9], suggesting a mechanism for altered KDM6A activity. However, we did not observe any differences in COMPASS complex components between WT and *Kmt2c* or *Kmt2d* KO cells in KDM6A immunoprecipitates (Extended Data Fig. 6i).

Finally, we assessed the therapeutic potential of KDM6A inhibition to prevent brain metastases. We pre-treated cells and mice with GSK-J4 for 2 d, followed by intracardiac injection and continued GSK-J4 treatment for 12 d. KDM6A inhibitor treatment was well tolerated and we did not observe any obvious alterations in vital organs such as the liver, intestine, spleen or kidneys (Extended Data Fig. 7a,b). However, we found that KDM6A inhibition significantly reduced *Kmt2c* KO cells in the brain, with a similar trend for *Kmt2d* KO cells (Fig. 6f and Extended Data Fig. 7c), whereas it did not affect the incidence of bone metastases (Extended Data Fig. 7d). We then repeated intracardiac injection of cells pre-treated with GSK-J4 for 2 d and harvested the brains 2 d later (Extended Data Fig. 7e). We detected significantly fewer *Kmt2c* KO cells in the brain with a similar trend observed for *Kmt2d* KO, suggesting an effect at early metastatic colonization (Extended Data Fig. 7f). Taken together, these results show that indirect inhibition of MMP3 via targeting KDM6A prevents metastasis of *KMT2C* or *KMT2D* mutant tumours (Fig. 4g).

## Discussion

Perturbed epigenetic programs due to mutations in histone-modifying enzymes, including *KMT2C*, can promote metastasis[10,11]. However, the underlying mechanisms are poorly understood, creating a major obstacle for therapeutic targeting of epigenetic drivers. Loss of both *Kmt2c* and *Kmt2d* can induce EMT or alter the EMT balance, which has been associated with metastasis[10,11]. However, we observed metastasis-promoting effects even in the absence of EMT.

*KMT2C* has been shown to affect DNA damage repair[20], whereas *KMT2D* is also commonly mutated in human mismatch repair-deficient tumours[21] with a superior response to ICI[14]. Additionally, *KMT2D* was identified as a determinant of the response to ICI. Consistently, we observed that *Kmt2c* or *Kmt2d* KO tumours have evidence of both immune activation and suppression.

Differential chromatin binding of monomethylated H3K4 in *Kmt2c* and *Kmt2d* KO cells and changes in H3K27 modifications and KDM6A binding imply involvement of other COMPASS complex components leading to broader epigenetic alterations. Indeed, in *Drosophila*, the KDM6A-interacting domain of Trr (KMT2C/KMT2D) was sufficient to rescue Trr-null lethality by stabilizing Utx (KDM6A)[7]. However, this might only happen in double mutants, which frequently occur in bladder or colorectal cancer but not in other cancer types[22]. Loss of KDM6A can increase KMT2D chromatin binding, implying a rescue function despite their distinct enzymatic function[23]. In line with this,

our data showed increased KDM6A binding upon *Kmt2c* or *Kmt2d* loss. This might also explain our finding of increased H3K4me1 and H3K27ac in contrast with previous publications[24,25]. H3K4me1 and H3K27me3 inversely correlate in a KDM6A-dependent manner, suggesting an indirect impact of KDM6A on H3K4me1 regulation[26]. Higher bimodal H3K4me1 signal at promoters correlates with increased gene activity[27], suggesting that H3K4me1 is also indirectly regulated upon *Kmt2c* and *Kmt2d* KO.

MMP3 has been shown to promote mammary tumorigenesis and has been proposed as a biomarker of cancer risk and tumour progression in patients with breast cancer[28]. An unanswered question in our study is whether MMP3 only plays a role in metastatic seeding or if its expression is also sustained in established metastases. MMP family members are important in tumour growth and metastasis, but MMP inhibitors have not been successful in clinical trials, in part due to their toxicity[29]. However, our study provides an alternative approach for targeting MMP3 via KDM6A inhibition to prevent brain metastases of *KMT2C* or *KMT2D* mutant tumours. As KDM6A inhibition seems to be well tolerated in mice, it could potentially be used in patients with TNBC who have *KMT2C* or *KMT2D* mutant tumours, in combination with currently used therapies. Alternative strategies are combinations including P300 and bromodomain and extra-terminal protein inhibitors, due to gained H3K27ac in *Kmt2c/Kmt2d* KO cells. Bromodomain and extra-terminal protein inhibitors have been shown to be more effective in *Kmt2c* KO cancer cells[11].

A remaining question is why KDM6A chromatin binding is enhanced upon *Kmt2c* or *Kmt2d* KO. Less is known about the redundancy of the COMPASS complexes and whether loss of either can rescue or even increase the activity of the other. It is likely that other COMPASS components beyond KDM6A also play a role in *Kmt2c* or *Kmt2d* loss-induced epigenetic and phenotypic changes. Our multi-dimensional profiling data provide a rich resource for further investigations into these unexplored areas.

## Online content

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

## Methods

### Ethical compliance
This research complies with all of the ethical and safety rules. Animal work was performed following protocol 11-023, approved by the Dana-Farber Cancer Institute Animal Care and Use Committee.

### Mammary tumour cell lines
168FARN and 67NR murine mammary tumour cell lines were obtained from the Karmanos Cancer Institute. HEK293T (CRL-3216) cells were obtained from the American Type Culture Collection and cultured following the provider's recommendations. Cell lines were regularly tested for *Mycoplasma*.

### Animal models
For MFP injection and intracardiac seeding, female BALB/c or NOD.Cg-Prkdcscid Il2rgtm1Wjl/SzJ (NSG) mice were purchased from The Jackson Laboratory at 5–6 weeks of age. Animal experiments were performed according to protocol 11-023 approved by the Dana-Farber Cancer Institute Animal Care and Use Committee. The maximum tumour size limit was 2 cm in any dimension and was not exceeded in this study. Mice were housed five to a cage with ad libitum access to food and water under an ambient temperature of 20 °C with a humidity of 40–50% and a 12 h light/12 h dark cycle.

### Cell line models
Single-guide RNA sequences (Supplementary Table 5) against exon 1 of murine *Kmt2c* or *Kmt2d* were chosen using University of California Santa Cruz Genome Browser CRISPR target tracks with a Doench/Fusi 2016 efficiency of >55. Three guide sequences each were then cloned into an sgRNA scaffold mCherry-tagged vector using BsmBI. Target cells were co-transfected with three sgRNA vectors and Cas9–GFP vector in a 1:1:1:3 ratio using Lipofectamine 3000 (Invitrogen) according to the manufacturer's protocol. After 2 d, mCherry/GFP double-positive cells were sorted via flow cytometry and cultured for 8 d until transient expression of the fluorescence marker was lost. The process was repeated two additional times. Finally, cells were sorted for mCherry/GFP double negativity to select out cells with random stable integration of either of the vectors. To stably integrate H2B–mCherry or shRNA vectors for knockdown of *Mmp3* or *Kdm6a*, HEK293T cells were transfected with packaging plasmids (pMD2.G and psPAX2) and respective vectors in a 1:2:2 ratio using Lipofectamine 3000 (Invitrogen). After 2 d, the supernatant was harvested and filtered through 0.22 μm filters, then 200 μl was used to infect target cells with the addition of polybrene (4 μg ml⁻¹) for 2 d. Cells were then either selected using puromycin (2 μg ml⁻¹) or sorted for mCherry positivity. For knockdown of *Mmp3* and *Kdm6a*, three different SMARTvector Lentiviral shRNA constructs incorporating the mEF1α promoter but no fluorescence marker (Horizon Discovery) were used.

### Mouse tumorigenesis and metastasis assays
For MFP or intracardiac injections, mice were anaesthetized by constant isoflurane application. Instruments and injection sites were sterilized, and for MFP injection a 1 cm incision was made between the midline and the fourth nipple. Using insulin syringes, 200,000 cells in 50 μl phosphate-buffered saline (PBS) were injected into the MFP and the incision was closed with surgical clips. Post-operational analgesia was ensured with topical ropivacaine application. For intracardiac injection, 50,000 cells in 50 μl PBS were injected directly into the left ventricle. Successful injection was validated by controlling blood flow into the syringe before and after the procedure. For KDM6A inhibition experiments, mice were treated with GSK-J4 HCl (Selleckchem; 10 mg kg⁻¹ in PBS) via intraperitoneal injection every other day, starting 2 d before intracardiac injection and lasting until the experimental endpoint.

### Cell proliferation and viability assays
Cells were cultured at 37 °C under 5% $CO_2$ and humidified conditions. For proliferation or dose curve analyses, 500 cells were plated in 96-well plates and treated with the indicated drugs in dimethyl sulfoxide (to a final concentration of 0.1%) or left untreated. At 5 d after treatment, the start cell count was measured with a Celigo Image Cytometer (Nexcelom Bioscience) using the mCherry signal of the cells. For proliferation analyses, the cell count was analysed every day for 5 d.

### Immunoblot analyses
Cells were collected and proteins were isolated using isolation buffer (1 mM ethylene glycol tetraacetic acid, 1 mM ethylenediaminetetraacetic acid (EDTA), 150 mM NaCl, 20 mM Tris and 1% Triton-X) for 30 min on ice after passing through a QIAshredder (QIAGEN). Histones were isolated using the EpiQuik Total Histone Extraction Kit (EpigenTek). The protein concentration was measured using Pierce 660 nm Protein Assay Reagent (Thermo Fisher Scientific) and equal amounts were loaded onto 3–8% gradient Tris-Acetate or 10% Tricine (histone extracts) gels. Proteins were transferred onto polyvinylidene difluoride membranes using wet blot devices and transfer buffer containing either 20% (histone extracts) or 10% (other extracts) methanol for 1 h (histone extracts) or 4 h (other extracts) at 90 V. Membranes were incubated with the primary antibodies (Supplementary Table 5) KMT2C (a gift from A. Shilatifard; 1:1,000), KMT2D (orb18454; Biorbyt; 1:1,000), tubulin (T6199; Millipore; 1:20,000), total histone H3 (39736; Active Motif; 1:1,000), H3K4me1 (710795; Invitrogen; 1:1,000), H3K27me3 (39155; Active Motif; 1:1,000), H3K27ac (ab4729; Abcam; 1:1,000), PAX-interacting protein 1 (ABE1877; Sigma–Aldrich; 1:1,000), RBBP5 (13171S; Cell Signaling Technology; 1:1,000) or WDR5 (13105S; Cell Signaling Technology; 1:1,000) in 5% milk overnight and with the secondary antibodies anti-mouse HRP (62-6520; Invitrogen; 1:10,000) or anti-rabbit HRP (65-6120; Invitrogen; 1:10,000) in Tris-buffered saline with 0.1% Tween-20 (TBST) for 1 h. Signals were developed with Clarity Western ECL Substrate (Bio-Rad) on a ChemiDoc MP device (Bio-Rad). If needed, antibodies were stripped off from membranes using Restore Western Blot Stripping Buffer (Thermo Fisher Scientific).

### Combined immunoprecipitation and immunoblot analyses
Approximately 15 million cells were collected and lysed in lysis buffer (300 mM NaCl, 50 mM Tris (pH 7.5), 1% IGEPAL CA-630 and 0.1% sodium deoxycholate) for 5 min in a cooled water bath sonicator and passed five times through a 30 G syringe. Next, 1.5 mg protein lysate was diluted up to 1 ml with low IP buffer from the Nuclear Complex Co-IP Kit (Active Motif), then 2.5 μg antibody (KDM6A; 33510S; Cell Signaling Technology) was added before incubation on a rotator at 4 °C. Following this, 50 μl Dynabeads Protein G was added for 2 h on a rotator at 4 °C. The beads were washed four times with washing buffer (150 mM NaCl and 50 mM Tris (pH 7.5)) and the proteins were eluted with 40 μl 4× LDS buffer at 95 °C for 5 min.

### Immunofluorescence staining
Tissue sections were deparaffinized and antigen retrieval was performed using Target Retrieval Solution, pH 6 (Agilent) for 40 min with a non-pressure food steamer. Slides were incubated with the primary antibodies mCherry (43590S; Cell Signaling Technology), CD8 (98941S; Cell Signaling Technology) and PDL-1 (64988S; Cell Signaling Technology) in 5% normal goat serum in TBST overnight and the secondary antibodies goat anti-rabbit Alexa Fluor 647 (A-21245; Invitrogen) and goat anti-rabbit Alexa Fluor 555 (A-21428; Invitrogen) in 5% normal goat serum in TBST for 2 h. Endogenous fluorescence was quenched using a TrueVIEW Autofluorescence Quenching Kit (Vector Laboratories) for 5 min. Images were taken with a Nikon ECLIPSE Ti2-E fluorescence microscope. Quantification was performed with automated scripts using Fiji (ImageJ) from three to five different fields of view per section.

Values within each replicate were averaged to calculate the final value per replicate.

## Flow cytometry
Immediately after collection, tissues were smashed with micro pestles and digested for 10 min (brain and bone marrow) or 1 h (tumour, liver and lungs) using digestion media (2% wt/vol collagenase IV, 2% wt/vol hyaluronidase and 2% wt/vol bovine serum albumin in Dulbecco's modified Eagle medium) at 37 °C on a shaker. Solutions were filtered through a mesh, washed with PBS and frozen in 10% dimethyl sulfoxide/foetal bovine serum at −80 °C or directly used for flow cytometry. For this, cells were passed through a 70 or 100 µm cell strainer, incubated with 4′,6-diamidino-2-phenylindol (1:20,000) and analysed using an LSRFortessa (BD Biosciences). Gating strategies can be found in Supplementary Fig. 1.

## Next-generation sequencing amplicon
Cells were washed with PBS and DNA was isolated using DNeasy Blood & Tissue Kits (QIAGEN). Regions of interest were amplified via PCR using primers for *Kmt2c* or *Kmt2d* (Supplementary Table 5) and run on an agarose gel electrophoresis. PCR products were purified from agarose gel using the Monarch DNA Gel Extraction Kit (New England Biolabs) and sequenced with GENEWIZ (Azenta).

## Quantitative real-time PCR
RNA was isolated using the Monarch Total RNA Miniprep Kit (New England Biolabs). Then, 2 µg RNA per reaction was used for reverse transcription with the PrimeScript RT Reagent Kit (Takara Bio). Real-time PCR was performed using TB Green Premix Ex Taq II (Takara Bio) on a CFX96 Touch Real-Time PCR Detection System (Bio-Rad). Values were calculated using the ΔΔCt method normalized to actin expression. Primer sequences for *β-actin*, *Mmp3*, *Mmp1b*, *Mmp10*, *Mmp13* and *Kdm6a* are indicated in Supplementary Table 5.

## Cytokine array
Cells were washed with PBS and cultured in foetal bovine serum-free Opti-MEM (Gibco) for 24 h before the supernatant was collected. Snap-frozen tumour tissue was mechanically homogenized and lysed with 1% Triton-X in PBS and the supernatant was collected. For each experimental condition, the protein concentration in the supernatant from five biological replicates was measured using Pierce 660 nm Protein Assay Reagent (Thermo Fisher Scientific) and the cytokine concentration was determined with a Proteome Profiler Mouse XL Cytokine Array (R&D Systems) on a ChemiDoc MP (Bio-Rad) device.

## Histone mass spectrometry
Cells were washed twice with cold PBS scraped off the plates, collected via centrifugation and snap frozen at −80 °C. Histones were isolated via acidic extraction followed by trichloroacetic acid precipitation, as previously described[31]. Briefly, 10 µg histone extract of each sample was propionylated, desalted and digested with trypsin overnight. The generated peptides were propionylated, desalted and reconstituted for mass spectrometry analysis. A reference mixture of isotopically labelled synthetic peptides for histones H3 and H4 was added to each sample. Peptides were separated on a C18 column (EASY-nLC 1000; Thermo Fisher Scientific) and analysed by mass spectrometry using the parallel reaction monitoring method (Q Exactive Plus Orbitrap; Thermo Fisher Scientific). Chromatographic peak areas of endogenous (light; L) and synthetic standard (heavy; H) peptides were extracted in Skyline and the ratios of light to heavy peak areas (L:H) were calculated. Ratios were normalized to respective ratios of a typically unmodified region of H3 (41–49) or H4 (68–78) and were log$_2$ transformed. *Kmt2c* KO and *Kmt2d* KO cells were further normalized to the WT for each histone mark.

## Single-cell RNA-seq preparation and sequencing
Samples from tumour tissues were prepared as described for flow cytometry. To deplete dead cells and debris, Percoll purification was performed. First, layers of 50 and 40% Percoll solution (90% Percoll and 10 mM HEPES in PBS) were prepared in 15 ml conical centrifugation tubes. Cells were pelleted, resuspended in 20% Percoll solution and layered on top of the former layers. Samples were centrifuged for 30 min at 2,000g and 4 °C without a break. Then, white layer interphases between 20 and 40% layers were collected, washed with PBS, filtered through a 100 µm strainer and pelleted. Pellets were resuspended in 0.04% UltraPure BSA (Sigma–Aldrich) in PBS and immediately processed for library preparations. Approximately 26,000 single cells were loaded onto a 10x Genomics Chromium instrument (10x Genomics) according to the manufacturer's recommendations. The scRNA-seq libraries were generated using a Chromium Next GEM Single Cell 5′ HT v2 Kit (10x Genomics). Quality controls for amplified complementary DNA libraries and final sequencing libraries were performed using a Bioanalyzer High Sensitivity DNA Kit (Agilent). Equimolar ratios of libraries were sequenced on an Illumina NovaSeq 6000 (Illumina) targeting 40 million 150-base pair (bp) read pairs per library at the Dana-Farber Cancer Institute Molecular Biology Core Facilities.

## Single-cell RNA-seq analyses
Raw data were processed using Cell Ranger (10x Genomics) with the bcl2fastq function to obtain fastq files for each sample. Files were further processed using Cell Ranger Count 7.0.1 and Cell Ranger Aggr version 7.0.1 (10x Genomics) to obtain counts and aggregate samples into groups. Countmatrix files were then processed in R studio using the Seurat package. Low-quality reads were removed according to the following criteria: nFeature_RNA > 500 & nFeature_RNA < 7500 & percent.mt < 15. Normalization was performed using LogNormalize followed by SCTransform for regression of the mitochondrial percentage and cell cycle (S.Score and G2M.Score; Supplementary Table 1). Clustering was performed using a resolution of 0.2 with the top 30 principal components. Next, clusters were manually annotated using gene.module.scores with specific marker genes (Supplementary Table 1). For further characterization, the clusters tumour cells and non-tumour cells were first subset. Non-tumour cells were then subset into the clusters fibroblast, endothelial cells, macrophage and T cell. Within each subset, cells were again annotated using gene.module.scores, and scores were plotted according to each genotype.

## RNA-seq preparation and sequencing
Cells were washed with PBS and collected via trypsinization. RNA was isolated using the RNeasy Mini Kit (QIAGEN). Libraries were prepared using KAPA mRNA HyperPrep (Roche) strand-specific sample preparation kits from 100 ng purified total RNA, according to the manufacturer's protocol, on a Biomek i7 (Beckman Coulter). The finished double-stranded DNA libraries were quantified by Qubit fluorometer (Thermo Fisher Scientific) and 4200 TapeStation (Agilent). Uniquely dual-indexed libraries were pooled in an equimolar ratio and shallowly sequenced on an Illumina MiSeq (Illumina) to further evaluate library quality and pool balance. The final pool was sequenced on an Illumina NovaSeq 6000 (Illumina) targeting 40 million 150-bp read pairs per library at the Dana-Farber Cancer Institute Molecular Biology Core Facilities.

## RNA-seq analyses
RNA-seq data were analysed using the VIPER[32] pipeline. Briefly, reads were aligned using STAR to the mm9 mouse genome. Genes with no counts in all samples were excluded and the remaining counts were normalized via log$_2$-transformed trimmed mean of *M* values transformation to counts per million (log$_2$[TMM−CPM + 1] from edgeR)[33] for further processing. To plot the heatmaps, batch effects among different biological replicates were removed using the removeBatchEffect

function of the LIMMA[34] package, considering each collection time as an individual batch, and z scores normalized to the average of all WT samples were plotted. For differential expression analyses, DESeq2 (ref. 35) was performed using genotypes as factor levels using non-normalized counts.

## ChIP-seq library preparation and sequencing
Cells were washed with cold PBS and fixed with fixing buffer (1% paraformaldehyde, 0.05 M HEPES (pH 7.5), 0.1 M NaCl and 1 µM EDTA (pH 8.0)) for 10 min at room temperature on a shaker. Fixation was stopped by adding 1/10 vol/vol 1.25 M glycine for 5 min. Cells were scraped off the plate, pelleted via centrifugation and stored at −80 °C. Then, cell membranes were lysed using lysis buffer (0.25% Triton-X, 0.5% IGEPAL CA-630, 10% glycerol, 0.5 µM EDTA, 0.14 M NaCl and 0.05 M HEPES (pH 8.0)) for 10 min at 4 °C and nuclei preparations were collected via centrifugation. Nuclei were washed with washing buffer (0.5 µM EDTA, 0.2 M NaCl and 0.01 M Tris-HCl) for 10 min at 4 °C, washed again and subsequently resuspended in shearing buffer (0.01 M Tris-HCl, 0.5 µM EDTA and 0.1% sodium dodecyl sulfate (SDS)). The solution was transferred into AFA Fiber tubes and sonicated with a Covaris E220 ultrasonicator (Covaris; peak incident power = 150 W, duty cycles = 5%; cycles per burst = 200) for 15 min at 4 °C. Then, debris were pelleted and the supernatant was transferred to new reaction tubes before the addition of Triton-X (1%), NaCl (0.14 M) and pre-washed Dynabeads Protein G (Thermo Fisher Scientific) and incubation for 1 h at 4 °C on a rotator. The supernatant was collected and the antibodies H3K4me1 (ab8895; Abcam; 2.5 µg per ChIP), H3K27me3 (9733S; Cell Signaling Technology; 2.5 µg per ChIP), H3K27ac (C15410196; Diagenode; 2.5 µg per ChIP), KDM6A (33510S; Cell Signaling Technology; 2.5 µg per ChIP) or P300 (ab275378; Abcam; 2.5 µg per ChIP) were added before incubation at 4 °C on a rotator overnight. The next day, pre-washed Dynabeads Protein G were added before incubation at 4 °C on a rotator for 2 h. The supernatant was removed and the beads were washed with low salt wash buffer (2 mM EDTA, 1% Triton-X, 0.1% SDS, 0.15 M NaCl and 0.02 M Tris-HCl (pH 8.0)), high salt wash buffer (2 mM EDTA, 1% Triton-X, 0.1% SDS, 0.5 M NaCl and 0.02 M Tris-HCl (pH 8.0)) and LiCl wash buffer (1 mM EDTA, 1% sodium deoxycholate, 1% IGEPAL CA-630, 0.25 M LiCl and 0.01 M Tris-HCl (pH 8.0)) at 4 °C for 5 min each. Beads were washed twice with TE buffer, and elution buffer (1% SDS and 100 mM NaHCO₃) was added and incubated for 30 min while being vortexed every 5 min at room temperature. The beads were removed and the supernatant was incubated at 65 °C overnight. Then, 0.2 mg ml⁻¹ RNase A was added before incubation for 30 min at 37 °C. Subsequently, 0.2 mg ml⁻¹ proteinase K was added before further incubation for 1 h at 55 °C. DNA was extracted by adding 1:1 vol/vol phenol/chloroform (pH 8.0) and the upper phase was used for DNA precipitation by adding isopropanol (42.5%), glycogen (0.007%) and sodium perchlorate (0.28 M), then pelleted using centrifugation. The DNA pellet was washed with 70% ethanol and resuspended in low TE buffer. ChIP-seq libraries were prepared using xGen DNA Library Prep reagents (Integrated DNA Technologies) on a Biomek i7 (Beckman Coulter) liquid-handling platform from approximately 1 ng DNA with 14 cycles of PCR amplification, according to the manufacturer's protocol. Finished sequencing libraries were quantified by Qubit fluorometer (Thermo Fisher Scientific) and 2200 TapeStation (Agilent). Library pooling and indexing was evaluated with shallow sequencing on an Illumina MiSeq (Illumina). Subsequently, libraries were sequenced on an Illumina NovaSeq 6000 (Illumina) targeting 40 million 150-bp read pairs by the Molecular Biology Core Facilities at the Dana-Farber Cancer Institute.

## ChIP-seq analyses
ChIP-seq data were analysed using the CoBRA[36] pipeline. In brief, reads were aligned to the mm9 genome using the BWA-MEM[37] aligner and peaks were called using MACS2 (ref. 38) (false discovery rate < 0.01;

broad peak mode for H3K4me1, H3K27me3, KDM6A and P300; default for H3K27ac). Reads in peaks were then counted using SAMtools[39] and used for differential analysis with DESeq2 (ref. 35) (P value < 0.05; log₂[fold change] > 0.5) normalized to sequencing depth. Heatmaps were generated using individual BigWig files and differential peak files for each comparison were generated using computeMatrix and plotHeatmap excluding blacklisted regions. The intersection of differential peaks was identified using Bedtools[40] with at least a 1-bp overlap. Motif analysis for each differential peakset was performed using HOMER[41] (-size 200 -p 10), and de novo motifs were considered. For motif heatmaps, only motifs with a −log₁₀[P value] > 25 in at least one sample were plotted. Correlation of differential peaks and gene expression was performed using BETA[17]. Each individual differential gained or lost peakset was analysed together with the DESeq2 output from RNA-seq of the same cells but different biological replicates using default parameters. For quantification of P300 signal intensities within H3K27ac peaks, multiBigwigSummary was used to bin the P300 signal of each biological replicate into specific peak regions. The bin count of replicates was averaged and log₂ transformed. Density plots for the correlation of changed P300 and H3K27ac signals was generated with ggplot. DESeq2 outputs from the CoBRA pipeline were used to extract all of the calculated fold changes within identified peak regions for H3K27ac or P300 in Kmt2c or Kmt2d KO cells compared with WT cells. Then, multiBigwigSummary[42] was used with a bin size of 100 to calculate the fold changes of H3K27ac or P300 in a similar region. Quantification of read counts for H3K27me3, P300 and KDM6A within the Mmp3 promoter (transcription start site ±1 kilobases) was performed using featureCounts[43] with respective bam files. For comparisons of signal intensities across the Mmp3 cluster, the locus signal of biological replicates was first averaged using bigwigAverage[42]. Then, average values were compared using bigwigCompare[42] in a 1,000-bp bin with the log2FC mode. The overall signal intensity was then calculated from average values of all of the bins.

## Analyses of public data
Mutation frequencies, locations at protein sequences and gene expression were identified using the cBioPortal platform (https://www.cbioportal.org/) selecting for patients with TNBC (oestrogen receptor, progesterone receptor and HER2 status negative) from the Molecular Taxonomy of Breast Cancer International Consortium cohort. This cohort consists of 320 patients, with 299 samples having mutation information and 320 samples having gene expression information. Gene expression for matched primary and secondary tumours was extracted from ref. 30 with selection for basal subtype. Plots for the correlation of metastatic potential scores and gene expression for cell lines were generated using data from the DepMap database (https://depmap.org/portal/).

## Statistics and reproducibility
Experiments and sample size were designed and determined according to similar experiments in previous studies. No statistical method was used to predetermine the required sample size. All repeat experiments are included in the analyses as biological replicates. Only animals with technical failures of cardiac injection were excluded. Metastasis quantification was performed after randomization of the samples. The investigators were not blinded to allocation during the experiments and outcome assessment. Data were first tested for normal distribution using the Shapiro−Wilk test. If data for matched comparisons did not pass the test, non-parametric tests were used. For single comparisons, a non-paired (unless otherwise noted), two-tailed t-test or Mann−Whitney U-test was used. For multiple comparisons, one-way analysis of variance corrected for multiple comparisons or a Kruskal−Wallis test corrected for multiple comparison was used. Unless otherwise noted, multiple comparison was performed for Kmt2c KO and Kmt2d KO versus the WT. All tests were performed

with a 95% confidence interval. *P* values are indicated for each experiment.

**Reporting summary**

Further information on research design is available in the Nature Portfolio Reporting Summary linked to this article.

## Data availability

All of the data needed to evaluate the conclusions in the paper are present in the paper and/or its Supplementary Information. RNA-seq, scRNA-seq and ChIP-seq data that support the findings of this study have been deposited in the Gene Expression Omnibus under the accession code GSE237392. The human TNBC data were derived from the Molecular Taxonomy of Breast Cancer International Consortium cohort. The dataset derived from this resource that supports the findings of this study is available at https://www.cbioportal.org/study/summary?id=brca_metabric. Mass spectrometry data have been deposited in ProteomeXchange with the primary accession code PXD052075. The mm9 mouse genome dataset is available at https://www.ncbi.nlm.nih.gov/datasets/genome/GCF_000001635.18/. Source data are provided with this paper. All of the other data supporting the findings of this study are available from the corresponding author upon reasonable request.

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

## Acknowledgements

We thank members of our laboratories for critical reading of this manuscript and useful discussions. We thank A. Shilatifard (Northwestern University) for providing KMT2C antibodies and S. Spisak (Dana-Farber Cancer Institute) for providing sgRNA/mCherry and Cas9/GFP plasmids. We thank the Dana-Farber Cancer Institute Molecular Biology and Flow Cytometry Core Facilities, Dana-Farber/Harvard Cancer Center Rodent Histopathology Core facility, Dana-Farber Cancer Institute Animal Resource Facilities and Translational Immunogenomics Laboratory for outstanding services. This research was supported by the National Cancer Institute (P01CA250959 to K.P. and H.W.L. and R35 CA197623 to K.P.), Ludwig Center at Harvard (to K.P.), Saverin Breast Cancer Research Fund (to K.P.), Canadian Institutes of Health Research (to M.-A.G.) and EMBO (to M.S.). The funders had no role in study design, data collection and analysis, decision to publish or preparation of the manuscript.

## Author contributions

M.S. and K.P. conceived of the study and wrote the original draft of the manuscript. M.S. developed the methodology. M.S., Z.L., A.H.R., M.P. and P.C. performed the formal analysis. M.S., Z.L., J.N., P.F., A.H.R., E.R.-J., M.-A.G., P.Y., S.R., P.C. and M.M.G. performed the investigation. H.W.L., M.P. and K.P. provided resources. H.W.L. and K.P. supervised the study and acquired funding. All authors contributed to designing the study, writing the manuscript and reviewing and editing the manuscript.

## Competing interests

K.P. serves on the Scientific Advisory Board of and holds equity options in IDEAYA Biosciences and Scorpion Therapeutics. K.P. and H.W.L. receive sponsored research funding through the Dana-Farber Cancer Institute from Novartis. The remaining authors declare no competing interests.

## Additional information

**Extended data** is available for this paper at https://doi.org/10.1038/s41556-024-01446-3.

**Correspondence and requests for materials** should be addressed to Kornelia Polyak.

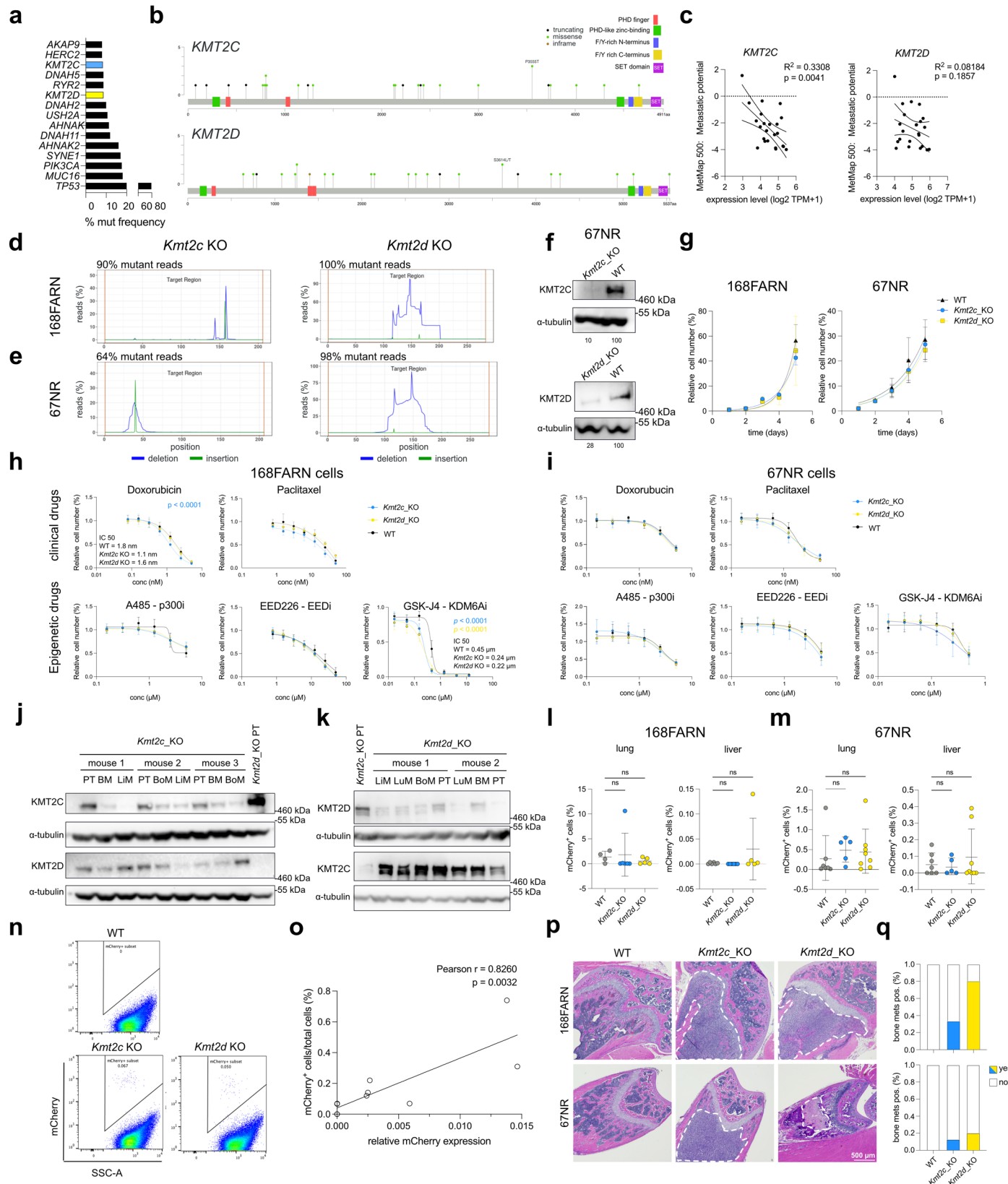

**Extended Data Fig. 1 | See next page for caption.**

**Extended Data Fig. 1 | *Kmt2c* or *Kmt2d* loss does not influence therapeutic responses but enables metastasis. a**, Top 15 most commonly mutated genes from TNBC breast cancer from METABRIC cohort (n = 299). **b**, Lolliplot plot of *KMT2C* or *KMT2D* mutation count. Generated with cBioPortal from data depicted in Ext. Data Fig. 1a. **c**, Correlation plot of log2 TPM+1 values for *KMT2C* and *KMT2D* expression with metastatic potential of breast cancer cell lines from DepMap (n = 23, simple linear regression, 95% confidence bands). **d** and **e**, Amplicon NGS sequencing for 168FARN (d) and 67NR (e) cells for *Kmt2c* KO or *Kmt2d* KO with percentage of deletions or insertion compared to WT sequence. **f**, Western blot for KMT2C or KMT2D from 67NR WT, *Kmt2c* KO or *Kmt2d* KO cells (n = 2 biological replicates). **g**, Growth curve analysis of 168FARN or 67NR WT, *Kmt2c* KO or *Kmt2d* KO cells (n = 3 biological replicates per group, non-linear regression with F test testing for growth rate, mean ± s.d., two-tailed). **h,i** Growth curve analysis for 168FARN (**h**) or 67NR (**i**) WT, *Kmt2c* KO or *Kmt2d* KO cells treated with the indicated drugs for 5 days normalized to matched DMSO. Significant differences were only seen for doxorubicin in 168FARN cells (1.8 nM for WT and 1.1 nM for *Kmt2c* KO) and GSK-J4 in 168FARN cells (0.45 μM for WT, 0.24 μM for *Kmt2c* KO and 0.22 μM for *Kmt2d* KO) (n = 6 biological replicates per group for 168FARN, n = 4 biological replicates per group (2 biological replicates for Doxorubicin) for 67NR, non-linear regression with F test testing for IC50, mean ± s.d., two-tailed).

**j,k** Western blot analysis for KMT2C, KMT2D and tubulin in indicated cell lines derived from primary tumours or distant organs from mice injected with *Kmt2c* KO (**j**) or *Kmt2d* KO (**k**) cells indicated in Fig. 1e. Cell line from *Kmt2d* KO (**j**) or *Kmt2c* KO (**k**) tumour was used as a control (n = 1 biological replicate). **l,m** Quantification of mCherry positive cells from lung or liver tissue of mice 12 days (**l**) or 13 days (**m**) after intracardiac injection with 168FARN (**l**) or 67NR (**m**) KO or WT cells (n = 5 (168FARN liver and lungs *Kmt2d* KO and 67NR liver and lungs *Kmt2c* KO), 6 (168FARN liver and lung WT and *Kmt2c* KO), 7 (67NR liver and lung WT) or 8 (67NR liver and lung *Kmt2d* KO) mice per group, Kruskal Wallis test with Dunn's multiple comparison, mean ± s.d.). **n**, Representative scatter plot from flow cytometry for mCherry using brain tissue from mice depicted in Fig. 1h. **o**, Correlation plot of mCherry positive cells and relative *mCherry* mRNA expression from brain of mice 11 days after intracardiac injection of *Kmt2c* or *Kmt2d* KO cells (n = 10 biological replicates, simple linear regression with Pearson correlation, two-tailed). **p**, H&E staining from femur sections of mice from experiment depicted in Fig. 1h and i (scale bar = 500 μm). **q**, Quantification of mice with or without bone metastases from experiments depicted in Fig. 1h and i (n = 5-8 mice per group). PT, primary tumour, BoM, bone metastases, LiM, liver metastases, BM, brain metastases, LuM, lung metastases.

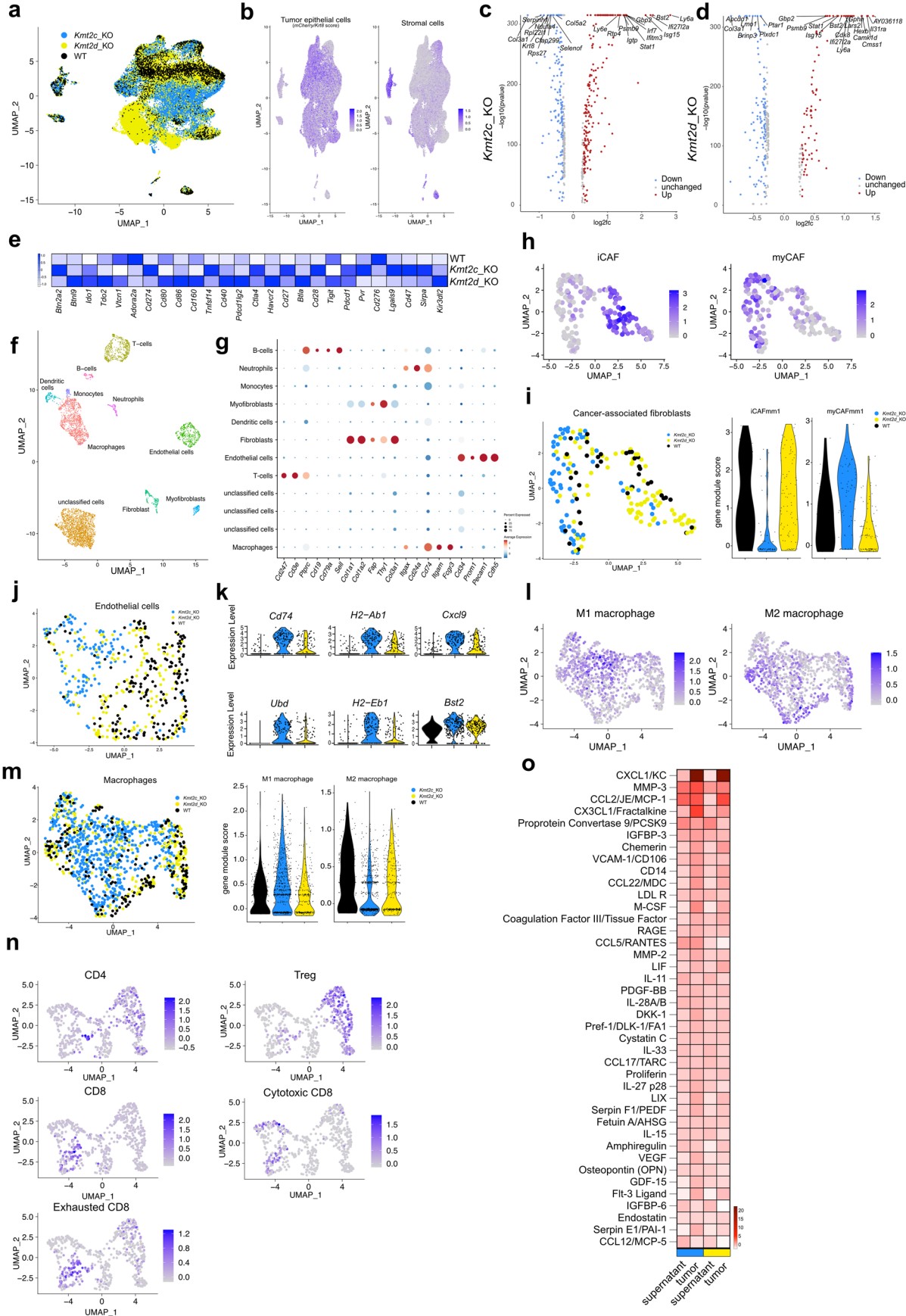

**Extended Data Fig. 2 | See next page for caption.**

**Extended Data Fig. 2 | Differential effects of *Kmt2c* KO or *Kmt2d* KO on distinct cell populations in tumour microenvironment. a**, UMAP from scRNA-seq with all cells coloured by genotype (n = 3 biological replicates per genotype). **b**, UMAP from scRNA-seq including all cells coloured by intensities of gene module scores for tumour (left) and non-tumour (right) gene signatures. **c**, Volcano plot showing significantly downregulated (blue) or upregulated (red) genes in *Kmt2c* KO cells compared to WT. DEGs are calculated using *findMarkers* (Wilcoxon Rank Sum test) from Seurat from subclustered tumour cells only and genes with log2 FC between −0.3 and 0.3 are not considered (grey). **d**, Volcano plot showing significantly downregulated (blue) or upregulated (red) genes in *Kmt2d* KO cells compared to WT. DEGs are calculated using *findMarkers* (Wilcoxon Rank Sum test) from Seurat from subclustered tumour cells only and genes with log2 FC between −0.3 and 0.3 are not considered (grey). **e**, Heatmap showing average expression of individual genes used to generate immune-checkpoint signature used in Fig. 2d. **f**, UMAP of all subclustered non-tumour cells coloured by annotations according to gene signatures. **g**, Dot plot showing expression levels of genes used to determine gene signatures for each annotation. **h**, UMAP of subclustered CAF populations coloured by intensities of gene signatures scores for iCAF (left) of myCAF (right). **i**, UMAP for subclustered CAFs coloured by genotype (left) and quantification of scores depicted in Ext. Data Fig. 2h for each genotype. **j**, UMAP of subclustered endothelial cells colored by different genotypes. **k**, Expression level of top 6 differentially upregulated genes in endothelial cells in *Kmt2c* KO, *Kmt2d* KO or WT tumours. **l**, UMAP of subclustered macrophage populations coloured by intensities for gene module scores for M1 (left) or M2 (right) macrophages. **m**, UMAP for subclustered macrophages coloured by genotype (left) and quantification of scores depicted in Ext. Data Fig. 2l for each genotype. **n**, UMAP for subclustered T cells coloured by intensities of gene signature scores for CD4, Treg, CD8, cytotoxic CD8 or exhausted CD8 cells. **o**, Heatmap showing log10 FC intensities from cytokine array using supernatant from cultured cells or tumour lysates from *Kmt2c* KO (blue) or *Kmt2d* KO (yellow) cells compared to WT cells (values are from 5 pooled samples per condition).

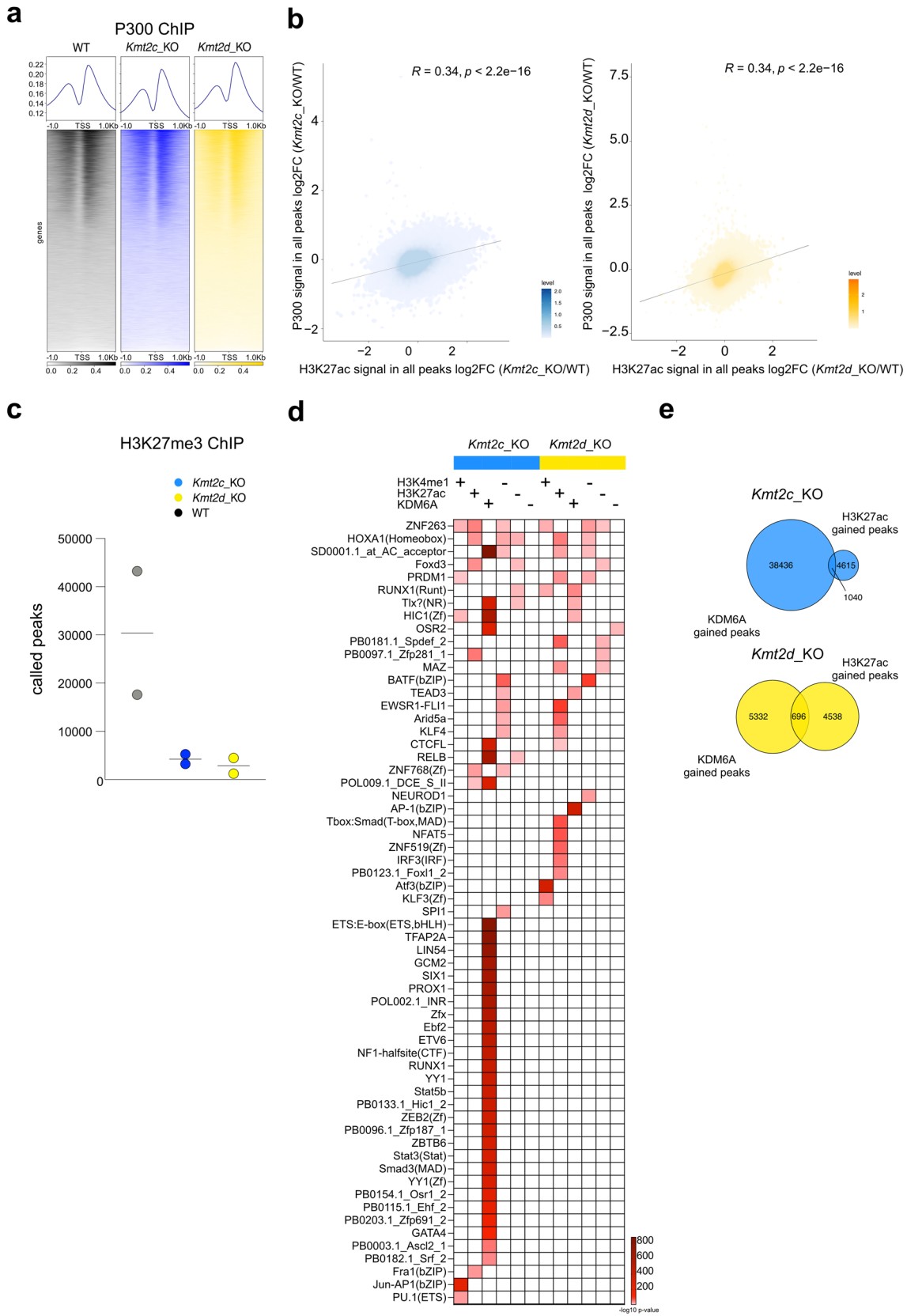

**Extended Data Fig. 3 | See next page for caption.**

**Extended Data Fig. 3 | Distinct histone modifier changes upon *Kmt2c* or *Kmt2d* KO. a**, Heatmap of total P300 ChIP-seq signal in WT, *Kmt2c*, and *Kmt2d* KO cells. Peaks are centred at TSS in a ± 3kb window (n = 2 biological replicates per group). **b**, Density plot showing log2FC in P300 and H3K27ac signal intensities in *Kmt2c* KO (left) and *Kmt2d* KO (right) cells compared to WT (linear regression with Pearson correlation, data is calculated from 3 biological replicates for H3K27ac and 2 biological replicates for P300 per group, two-tailed, p = 2.2 × $10^{-16}$). **c**, Quantification of total called peaks for H3K27me3 in WT, *Kmt2c* KO and *Kmt2d* KO cells (n = 2 biological replicates per group). **d**, Motif analysis using HOMER for significantly gained or lost peaks from H3K4me1, H3K27ac and KDM6A ChIP-seq in *Kmt2c* KO (blue) and *Kmt2d* KO (yellow) cells compared to WT cells. Data are presented as -log10 p-value. Only motifs with -log10 p-values > 25 in at least one sample are shown. **e**, Venn diagram showing intersections of significantly gained H3K27ac and KDM6A peaks for *Kmt2c* KO (blue) and *Kmt2d* (yellow) compared to WT cells.

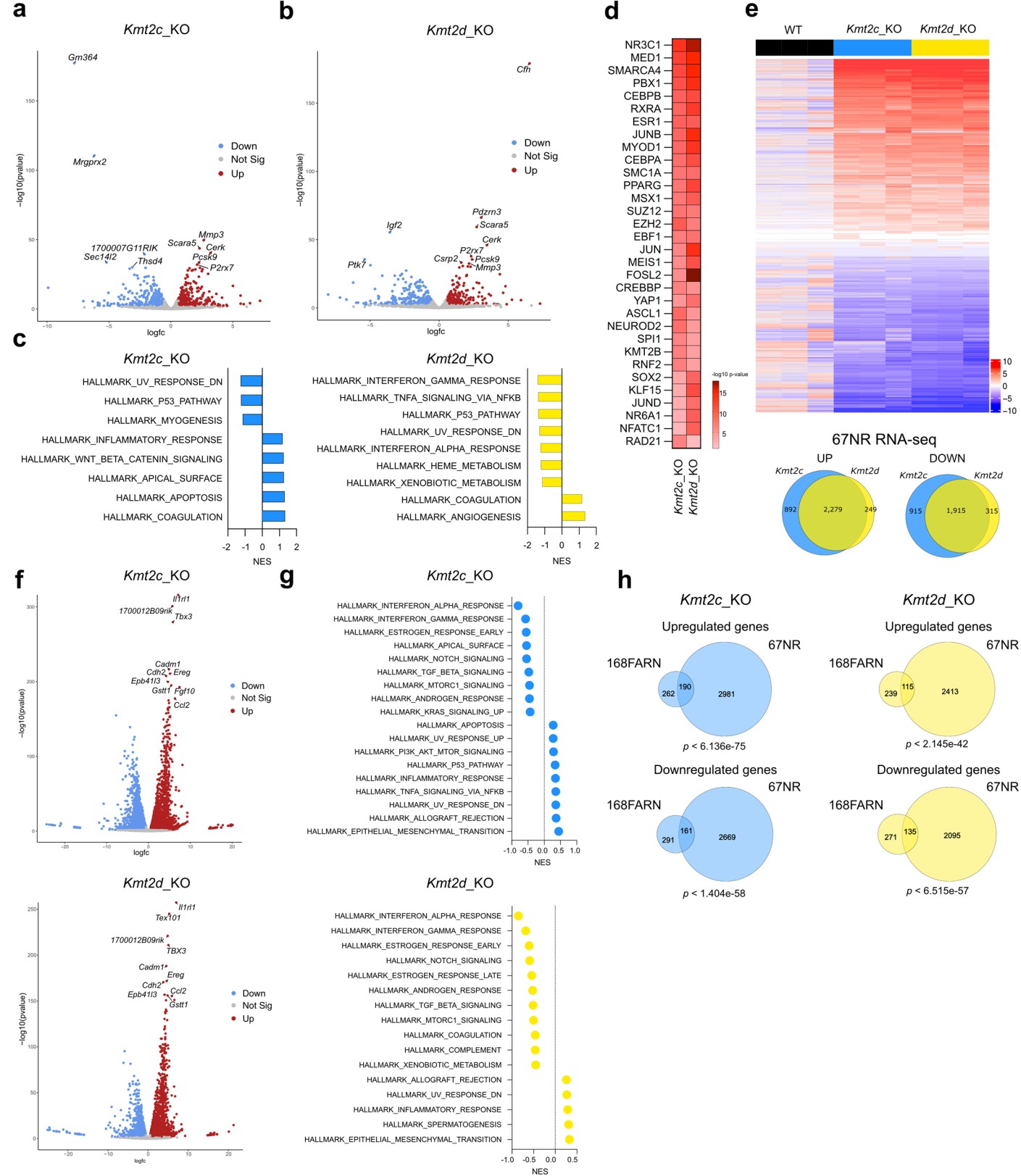

**Extended Data Fig. 4 | See next page for caption.**

**Extended Data Fig. 4 | *Kmt2c* and *Kmt2d* KO cell have unique but also shared transcriptomic landscapes. a**, Volcano plot showing significantly upregulated (red) and downregulated (blue) genes in *Kmt2c* KO cells compared to WT cells. Genes with adjusted p-values > 0.1 or log2 FC between −0.6 and 0.6 are not considered (grey). **b**, Volcano plot showing significantly upregulated (red) and downregulated (blue) genes in *Kmt2d* KO cells compared to WT cells. Genes with adjusted p-values > 0.1 or log2 FC between −0.6 and 0.6 are not considered (grey). **c**, GSEA analysis showing normalized enrichment scores for Hallmark gene sets of *Kmt2c* KO (left) or *Kmt2d* KO (right) cells compared to WT cells. Only pathways with p-values < 0.05 are shown. **d**, Transcription regulator prediction using LISA algorithm with upregulated DGE shown in Extended Data Fig. 3d,e. Shown are top 25 candidates for each KO cell line as -log10 (p-value). **e**, Heatmap (top) for RNA-seq data from WT (black), *Kmt2c* KO (blue) and *Kmt2d* KO (yellow) 67NR cells. Values are shown as z-scores normalized to average of WT cells. Only genes with adjusted p-values < 0.1 or log2 FC between −0.6 and 0.6 from DEG analysis are shown (n = 3 biological samples per group). Venn diagram (bottom) showing overlap of significantly upregulated (left) or downregulated (right) genes between *Kmt2c* KO (blue) and *Kmt2d* KO (yellow) cells compared to WT 67NR cells. **f**, Volcano plot showing significantly upregulated (red) and downregulated (blue) genes in *Kmt2c* KO (top) cells compared to WT 67NR cells. Genes with adjusted p-values > 0.1 or log2 FC between −0.6 and 0.6 are not considered (grey). Volcano plot showing significantly upregulated (red) and downregulated (blue) genes in *Kmt2d* KO (bottom) cells compared to WT 67NR cells. Genes with adjusted p-values > 0.1 or log2 FC between −0.6 and 0.6 are not considered (grey). **g**, GSEA analysis showing normalized enrichment scores for Hallmark gene sets of *Kmt2c* KO (top) or *Kmt2d* KO (bottom) cells compared to WT 67NR cells. Only pathways with p-values < 0.05 are shown. **h**, Venn diagram showing overlap of significantly upregulated (top) or downregulated (bottom) genes for *Kmt2c* KO (blue) or *Kmt2d* KO (yellow) between 168FARN and 67NR cells. P-values are calculated as exact hypergeometric probability of overlaps of each gene set normalized to total mouse transcriptome.

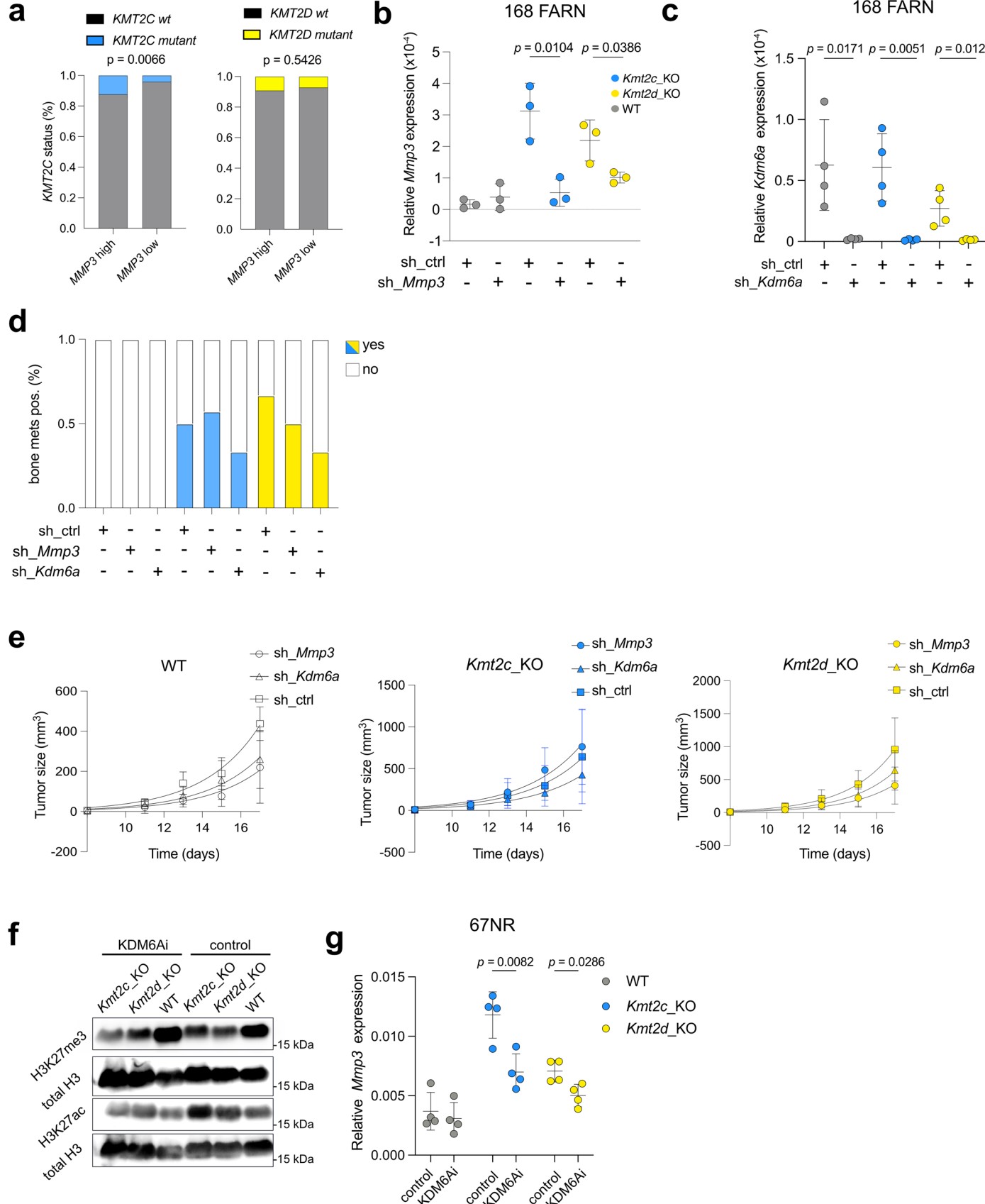

**Extended Data Fig. 5 | See next page for caption.**

**Extended Data Fig. 5 | *Mmp3* and *Kdm6a* in *Kmt2c* and *Kmt2d* KO-related tumorigenesis and metastases. a**, Relative mutation frequency of *KMT2C* (left) or *KMT2D* (right) in *MMP3* high (top 50%) or *MMP3* low (bottom 50%) TNBC from METABRIC cohort (n = 299, Fisher's exact test) **b**, Relative expression of *Mmp3* in control and sh_*Mmp3* knockdown derivatives of WT, *Kmt2c* KO and *Kmt2d* KO cells. Values were calculated using ddCt method normalized to actin expression (n = 3 biological replicates per group, two-tailed unpaired t-test, means ± s.d.). **c**, Relative *Kdm6a* mRNA expression in sh_ctrl and sh_*Kdm6a* derivatives of *Kmt2c* KO, *Kmt2d* KO and WT 168FARN cells measured by qRT-PCR (n = 4 biological replicates per group, two-tailed unpaired t-test, mean ± s.d.). **d**, Frequency of mice with bone metastases after intracardiac injection of WT, *Kmt2c* KO or *Kmt2d* KO cell derivatives with control shRNA or with shRNA against *Mmp3* or *Kdm6a*,

respectively (cohorts are the same as depicted in Figure 4f and i). **e**, Volumes of primary tumours after MFP injection of sh_*Mmp3*, sh_*Kdm6a* or sh_ctrl derivatives of WT (left), *Kmt2c* KO (middle) or *Kmt2d* KO (right) cells into BALB/c mice (n = 4 (WT sh_*Mmp3*), 5 (WT sh_ctrl and sh_*Kdm6a*, *Kmt2c* KO sh_sh_*Kdm6a* and *Kmt2d* KO sh_ctrl), 6 (*Kmt2d* KO sh_*Kdm6a* and sh_*Mmp3*) or 7 (*Kmt2c* KO sh_ctrl and sh_*Mmp3*) mice per group, non-linear regression with F test testing different growth rates, mean ± s.d., two-tailed). **f**, Western blot for H3K27me3, H3K27ac and total H3 in WT, *Kmt2c* KO and *Kmt2d* KO with or without GSK-J4 treatment for 2 days. (n = 2 biological replicates). **g**, Relative expression of *Mmp3* in DMSO or GSK-J4-treated WT, *Kmt2c* KO and *Kmt2d* KO 67NR cells 3 days after treatment start (n = 4 biological replicates per group, two-tailed Mann-Whitney test for WT and *Kmt2c* KO, unpaired t-test for *Kmt2d* KO, mean ± s.d.).

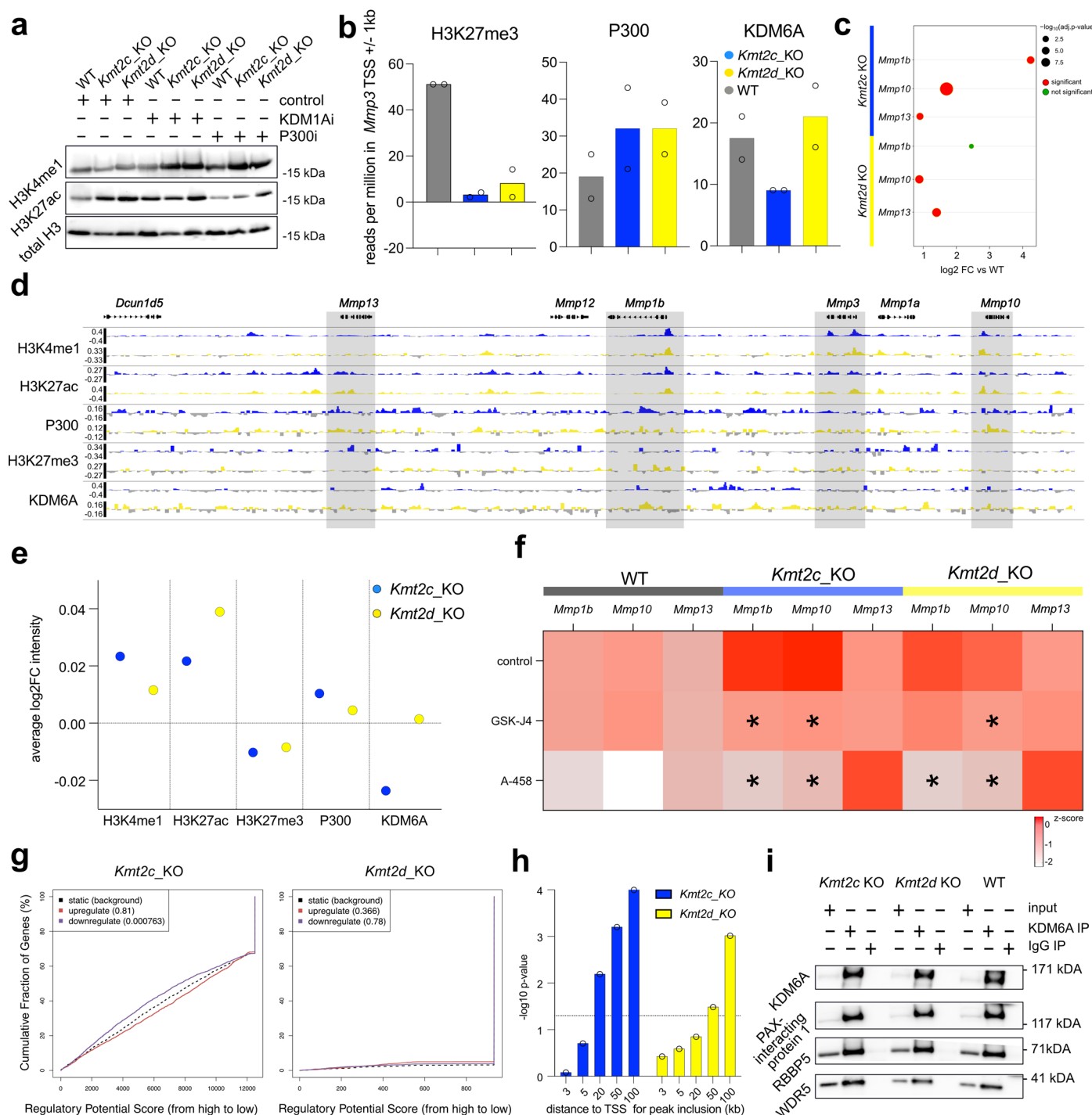

**Extended Data Fig. 6 | Regulation of *Mmp3* by KDM6A and P300 in *Kmt2c* and *Kmt2d* KO cells. a**, Representative western blot for H3K4me1, H3K27ac and total H3 proteins (n = 2 biological replicates) in WT, *Kmt2c* or *Kmt2d* KO cell treated with ORY-1001 (500 nM, KDM1A inhibitor) or A-485 (1μM, P300 inhibitor). **b**, Quantification of reads using featureCounts of H3K27me3, P300 and KDM6A ChIPseq data within ±1kb from the *Mmp3* promoter in WT, *Kmt2c* or *Kmt2d* KO cells (n = 2 biological replicates each). **c**, Bubble plot showing log2 fold change vs WT and -log10 adjusted p-value for *Mmp1b*, *Mmp10* and *Mmp13* expression in *Kmt2c* KO (top) or *Kmt2d* KO (bottom) from DEseq2 output using same data as Extended Data Fig 4a (n = 3 biological replicates each, Wald test). **d**, Gene tracks of log2FC of *Kmt2c* KO (blue) or *Kmt2d* KO (yellow) vs WT for H3K4me1, H3K27ac, P300, H3K27me3 and KDM6A across the *Mmp3* containing Mmp cluster locus. Scores were averaged in 1 kb windows to calculate fold changes (data are averaged from n = 3 biological replicates for H3K4me1 and H3K27ac or n = 2 biological replicates for H3K27me3, P300 and KDM6A). **e**, Quantification of average signal intensities for gene track signals in e in 1 kb bins. **f**, Relative

mRNA expression of *Mmp1b*, *Mmp10* and *Mmp13* in WT, *Kmt2c* or *Kmt2d* KO cells after control, GSK-J4 or A-485 treatment for 2 days. Values are plotted as z-scores after log₁₀ transformation for each gene (n = 4 biological replicates each, One-Way ANOVA with Dunnetts multiple comparison compared to control treatment within each genotype and gene, Kruskal-Wallis test with Dunns multiple comparison compared to control for *Kmt2d* KO cells and *Mmp13*). **g**, Correlation plots from BETA analysis using significantly gained KDM6A peaks within 3 kb of known TSS and differential gene expression in *Kmt2c* (left) or *Kmt2d* KO (right) cells compared to WT (data from n = 2 biological replicates for KDM6A ChIP-seq and n = 3 biological replicates for RNA-seq). **h**, Bar plot showing p-values for correlations of significantly gained KDM6A peaks within depicted distances from TSS with gene expression from BETA analysis (data from n = 2 biological replicates for KDM6A ChIP-seq and n = 3 biological replicates for RNA-seq). **i**, Immunoblot for KDM6A, PTIP, RBBP5 or WDR5 on control IgG or KDM6A immunoprecipitates (IP) using lysates of *Kmt2c* KO, *Kmt2d* KO or WT cells.

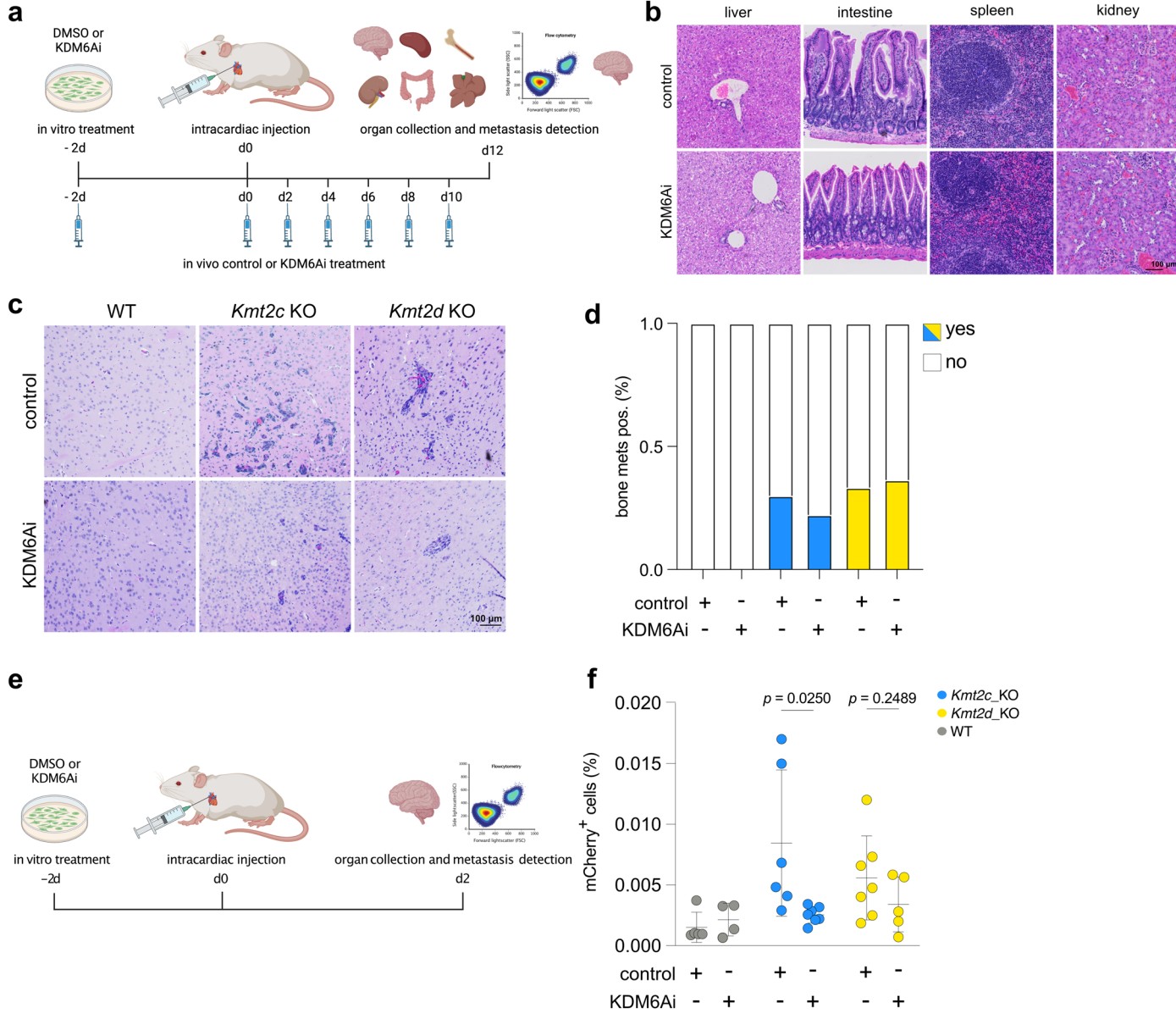

**Extended Data Fig. 7 | KDM6A inhibition is tolerable and inhibits brain metastases. a**, Schematic outline of experimental setup. **b**, Representative H&E-stained slides of liver, intestine, spleen and kidney from mice 12 days after intracardiac injection with WT cells treated with DMSO or GSK-J4. (scale bars = 100 μm) **c**, Representative H&E-stained slide from brain tissues of mice described in Fig. 4m. (Scale bars = 100 μm, representative data from 5 untreated and 4 treated mice from 1 experiment) **d**, Ratio of mice with detectable bone metastases from the mouse cohort described in Fig. 4m (n = 6–7 biological

replicates per group). **e**, Schematic outline of experimental setup for extended data figure 7f. **f**, Quantification of mCherry positive cells normalized to live cells in brain tissue lysates from mice. Mice were injected via intracardiac injection with WT, *Kmt2c* KO or *Kmt2d* KO cells which were treated with or without GSK-J4 for 2 days and brains were collected 2 days later (n = 4 (WT GSK-J4), 5 (WT DMSO and Kmt2d KO GSK-J4), 6 (Kmt2c KO DMSO) or 7 (Kmt2d KO DMSO and Kmt2c KO GSK-J4) mice per group, two-tailed Mann-Whitney test, mean and SD). Panels **a** and **e** created with BioRender.com.

# Reporting Summary

## Statistics

For all statistical analyses, confirm that the following items are present in the figure legend, table legend, main text, or Methods section.

| n/a | Confirmed | |
|---|---|---|
| ☐ | ☒ | The exact sample size (*n*) for each experimental group/condition, given as a discrete number and unit of measurement |
| ☐ | ☒ | A statement on whether measurements were taken from distinct samples or whether the same sample was measured repeatedly |
| ☐ | ☒ | The statistical test(s) used AND whether they are one- or two-sided *Only common tests should be described solely by name; describe more complex techniques in the Methods section.* |
| ☒ | ☐ | A description of all covariates tested |
| ☐ | ☒ | A description of any assumptions or corrections, such as tests of normality and adjustment for multiple comparisons |
| ☐ | ☒ | A full description of the statistical parameters including central tendency (e.g. means) or other basic estimates (e.g. regression coefficient) AND variation (e.g. standard deviation) or associated estimates of uncertainty (e.g. confidence intervals) |
| ☐ | ☒ | For null hypothesis testing, the test statistic (e.g. *F*, *t*, *r*) with confidence intervals, effect sizes, degrees of freedom and *P* value noted *Give P values as exact values whenever suitable.* |
| ☒ | ☐ | For Bayesian analysis, information on the choice of priors and Markov chain Monte Carlo settings |
| ☒ | ☐ | For hierarchical and complex designs, identification of the appropriate level for tests and full reporting of outcomes |
| ☒ | ☐ | Estimates of effect sizes (e.g. Cohen's *d*, Pearson's *r*), indicating how they were calculated |

*Our web collection on statistics for biologists contains articles on many of the points above.*

## Software and code

Policy information about availability of computer code

| | |
|---|---|
| Data collection | scRNAseq: About 26,000 single cells were loaded onto a 10x Genomics ChromiumTM instrument (10x Genomics) according to the manufacturer's recommendations. The scRNAseq libraries were generated using Chromium Next GEM Single Cell 5' HT Kit v2 (10x Genomics). Demultiplexing, barcoded processing, feature counting and aggregation was done using cellranger v7.0.1. ChIPseq: ChIPseq libraries were prepared using xGen DNA library prep reagents (IDT) on a Biomek i7 (Beckman Coulter) liquid handling platform from approximately 1ng of DNA with 14 cycles of PCR amplification according to manufacturer's protocol. Fastq data was processed using CoBRA pipeline (cobra 2.0, https://bitbucket.org/cfce/cobra/src/master/). Steps include alignment using BWA mem, filtering for uniquely mapped read and sorting bam files. Peaks were called using macs2 with default settings (-broad mode for H3K4me1 and H3K27me3). Output bedgraph files were then used to generate bigWig files. RNAseq: libraries were prepared from 100 ng RNA using Kapa mRNA HyperPrep (Roche) according to manufacturers protocol. VIPER pipeline (https://github.com/hanfeisun/viper-rnaseq/blob/master/cfce/README_CFCE.md) was used for alignment, assembly and gene counting. Flow cytometry: Single cells were acquired using a 4-laser and 17-parameter BD LSR Fortessa Cell Analyzer |
| Data analysis | scRNAseq: About 26,000 single cells were loaded onto a 10x Genomics ChromiumTM instrument (10x Genomics) according to the manufacturer's recommendations. The scRNAseq libraries were generated using Chromium gNext GEM Single Cell 5' HT Kit v2 (10x Genomics). Demultiplexing, barcoded processing, feature counting and aggregation was done using cellranger v7.0.1. ChIPseq: ChIPseq libraries were prepared using xGen DNA library prep reagents (IDT) on a Biomek i7 (Beckman Coulter) liquid handling platform from approximately 1ng of DNA with 14 cycles of PCR amplification according to manufacturer's protocol. Fastq data was processed using CoBRA pipeline (https://bitbucket.org/cfce/cobra/src/master/). Steps include alignment using BWA mem, filtering for uniquely mapped read and sorting bam files. Peaks were called using macs2 with default settings (-broad mode for H3K4me1 and H3K27me3). Output bedgraph files were then used to generate bigWig files. RNAseq: libraries were prepared from 100 ng RNA using Kapa mRNA HyperPrep (Roche) according to manufacturers protocol. VIPER pipeline |

(https://github.com/hanfeisun/viper-rnaseq/blob/master/cfce/README_CFCE.md) was used for alignment, assembly and gene counting.

R studio 2022.10.0-daily+67, seurat 4.3.0, DEseq2 1.38.3, GSEA 4.2.1, Graph Pad Prism 10, FlowJo 10.9.0, Fiji v1.54f

For manuscripts utilizing custom algorithms or software that are central to the research but not yet described in published literature, software must be made available to editors and reviewers. We strongly encourage code deposition in a community repository (e.g. GitHub). See the Nature Portfolio guidelines for submitting code & software for further information.

## Data

Policy information about availability of data

All manuscripts must include a data availability statement. This statement should provide the following information, where applicable:
- Accession codes, unique identifiers, or web links for publicly available datasets
- A description of any restrictions on data availability
- For clinical datasets or third party data, please ensure that the statement adheres to our policy

All data needed to evaluate the conclusions in the paper are present in the paper and/or the Supplemental Information. . All raw and processed genomic data was deposited to GEO under accession number: GSE237392.
Mass spectrometry data have been deposited in ProteomeXchange with the primary accession code PXD052075 https://proteomecentral.proteomexchange.org/cgi/GetDataset?ID=PXD052075.

## Research involving human participants, their data, or biological material

Policy information about studies with human participants or human data. See also policy information about sex, gender (identity/presentation), and sexual orientation and race, ethnicity and racism.

| Reporting on sex and gender | N/A |
|---|---|
| Reporting on race, ethnicity, or other socially relevant groupings | N/A |
| Population characteristics | N/A |
| Recruitment | N/A |
| Ethics oversight | N/A |

Note that full information on the approval of the study protocol must also be provided in the manuscript.

# Field-specific reporting

Please select the one below that is the best fit for your research. If you are not sure, read the appropriate sections before making your selection.

☒ Life sciences   ☐ Behavioural & social sciences   ☐ Ecological, evolutionary & environmental sciences

For a reference copy of the document with all sections, see nature.com/documents/nr-reporting-summary-flat.pdf

# Life sciences study design

All studies must disclose on these points even when the disclosure is negative.

| Sample size | Sample sizes for mouse experiments, cell culture and sequencing experiments were chosen according to former experiences with similar experiments to reach sufficient statistical power (see Shu et al. "Molecular cell (2020) or Janiszewska et al. Nature cell biology (2019)). |
|---|---|
| Data exclusions | For mouse metastases models, mice with failed intracardiac injection were excluded. Failed injection (wrong ventricle injection) was specified as the onset of high metastatic burden exclusively in the lung but no other tissue 4 - 5 days before collection of the remaining comparable group. For ChIP and scRNAseq low quality reads were excluded as described in the method and quality control section. |
| Replication | All in vitro and sequencing experiments were conducted with biological replicates (defined as 1 passaging (~5 days) between collection of cell line replicates). Experiments were replicated 2-4 times as indicated in the according figure legend. All replications were included in the data analysis. Mouse experiments were not individually replicated to comply with the 3R principle for animal research, however, sufficient samples sizes were chosen for each experiment. |
| Randomization | For in vivo studies mice were randomly allocated to injection of different cell lines. No other randomization was used. |
| Blinding | Sample collection for mouse experiments was not blinded, however, samples were randomized before flow cytometry and imaging and allocated to samples afterwards. Other experiments were not blinded as it would not affect results or interpretation. |

# Reporting for specific materials, systems and methods

We require information from authors about some types of materials, experimental systems and methods used in many studies. Here, indicate whether each material, system or method listed is relevant to your study. If you are not sure if a list item applies to your research, read the appropriate section before selecting a response.

## Materials & experimental systems

| n/a | Involved in the study |
|-----|----------------------|
| ☐ | ☒ Antibodies |
| ☐ | ☒ Eukaryotic cell lines |
| ☒ | ☐ Palaeontology and archaeology |
| ☐ | ☒ Animals and other organisms |
| ☒ | ☐ Clinical data |
| ☒ | ☐ Dual use research of concern |
| ☒ | ☐ Plants |

## Methods

| n/a | Involved in the study |
|-----|----------------------|
| ☐ | ☒ ChIP-seq |
| ☐ | ☒ Flow cytometry |
| ☒ | ☐ MRI-based neuroimaging |

## Antibodies

| Antibodies used | KMT2C rb 1/1000 gift from Ali Shilatifard  western blot<br>KMT2D rb 1/1000 Biorbyt orb184541 western blot<br>tubulin ms 1/20000 Millipore Sigma T6199 western blot<br>total histone H3 ms 1/1000 Active motif 39763 western blot<br>H3K4me1 rb 1/1000 Invitrogen 710795 western blot<br>H3K27me3 rb 1/1000 Active motif 39155 western blot<br>H3K27ac rb 1/1000 abcam ab4729 western blot<br>KDM6A rb 1/1000 Cell Signaling 33510S western blot<br>PAXIP1 rb 1/1000 Sigma Aldrich ABE1877 western blot<br>RBBP5 rb 1/1000 Cell Signaling 13171S western blot<br>WDR5 rb 1/1000 Cell Signaling 13105S western blot<br>KDM6A rb 2.5 ug/IP Cell Signaling 33510S immunoprecipitation<br>H3K4me1 rb 2.5 ug/ChIP abcam ab8895 ChIP<br>H3K27me3 rb 2.5 ug/ChIP Cell Signaling 9733S ChIP<br>H3K27ac rb 2.5 ug/ChIP Diagenode C15410196 ChIP<br>KDM6A rb 2.5 ug/ChIP Cell Signaling 33510S ChIP<br>P300 rb 2.5 ug/ChIP abcam ab275378 ChIP<br>CD8 rb 1/500 Cell Signaling 98941S immunofluorescence<br>PDL-1 rb 1/200 Cell Signaling 64988S immunofluorescence<br>mCherry rb 1/500 Cell Signaling 43590S immunofluorescence<br>rb anti goat-HRP 1/10000 Invitrogen 65-6120 western blot<br>ms anti goat-HRP 1/10000 Invitrogen 62-6520 western blot<br>rb anti goat Alexa Fluor 6471/500 Invitrogen A-21245 immunofluorescence<br>rb anti goat Alexa Fluor 555 1/500 Invitrogen A-21428 immunofluorescence |
|---|---|
| Validation | Validation of all the commercial antibodies can be found on the manufacturer's website using the provided catalog number. Antibodies for ChIPseq were validated using the analysis QC control.<br><br>Specific validation of antibodies:<br>KMT2C/KMT2D orb184541 -  validated with confirmed knockout samples in this manuscript<br>tubulin T6199 several enhanced validation approaches https://www.sigmaaldrich.com/US/en/product/sigma/t6199<br>total histone H3 39763 This antibody has been validated for use in ChIP and/or ChIP-Seq https://www.activemotif.com/catalog/details/39763<br>H3K4me1 710795 This Antibody was verified by Peptide array to ensure that the antibody binds to the antigen stated. https://www.thermofisher.com/antibody/product/H3K4me1-Antibody-Recombinant-Polyclonal/710795<br>H3K27me3 39155 This antibody has been validated for use in ChIP and/or ChIP-Seq https://www.activemotif.com/catalog/details/39155<br>H3K27ac ab4729 Suitable for: ICC/IF, WB, IHC-P, ChIP, PepArr https://www.abcam.com/products/primary-antibodies/histone-h3-acetyl-k27-antibody-chip-grade-ab4729.html<br>KDM6A 33510S validation via western blot of known positive and negative samples https://www.cellsignal.com/products/primary-antibodies/utx-d3q1i-rabbit-mab/33510<br>PAXIP1 ABE1877 Evaluated by Western Blotting with with recombinant Pax-interacting protein 1. https://www.emdmillipore.com/US/en/product/Anti-PAXIP1,MM_NF-ABE1877<br>RBBP5 13171S validation via immunoprecipitation https://www.cellsignal.com/products/primary-antibodies/rbbp5-d3i6p-rabbit-mab/13171<br>WDR5 13105S This antibody has been validated using SimpleChIP® Enzymatic Chromatin IP Kits. https://www.cellsignal.com/products/primary-antibodies/wdr5-d9e1i-rabbit-mab/13105<br>KDM6A 33510S validation via western blot of known positive and negative samples https://www.cellsignal.com/products/primary-antibodies/utx-d3q1i-rabbit-mab/33510<br>H3K4me1 ab8895 confirmed specificity through extensive validation https://www.abcam.com/products/primary-antibodies/histone- |

h3-mono-methyl-k4-antibody-chip-grade-ab8895.html
H3K27me3 9733S tested for specificity and cross reactions https://www.cellsignal.com/products/primary-antibodies/tri-methyl-histone-h3-lys27-c36b11-rabbit-mab/9733
H3K27ac C15410196 validation via immunoprecipitation, ChIP and cross-ractivity studies https://www.diagenode.com/en/p/h3k27ac-polyclonal-antibody-premium-50-mg-18-ml
KDM6A 33510S validation via western blot of known positive and negative samples https://www.cellsignal.com/products/primary-antibodies/utx-d3q1i-rabbit-mab/33510
P300 ab275378 validation via immunoprecipitation https://www.abcam.com/products/primary-antibodies/kat3b--p300-antibody-epr23495-268-chip-grade-ab275378.html
CD8 98941S validation via western blot of known positive and negative samples https://www.cellsignal.com/products/primary-antibodies/cd8a-d4w2z-xp-rabbit-mab/98941
PDL-1 64988S specificity has been validated in known samples in this study https://www.cellsignal.com/products/primary-antibodies/pd-l1-d5v3b-rabbit-mab/64988
mCherry 43590S validation via western blot of known positive and negative samples https://www.cellsignal.com/products/primary-antibodies/mcherry-e5d8f-rabbit-mab/43590
rb 65-6120 tested for specificity and cross reactions https://www.thermofisher.com/antibody/product/Goat-anti-Rabbit-IgG-H-L-Secondary-Antibody-Polyclonal/65-6120
ms 62-6520 tested for specificity and cross reactions https://www.thermofisher.com/antibody/product/Goat-anti-Mouse-IgG-H-L-Secondary-Antibody-Polyclonal/62-6520
rb A-21245 whole antibodies have been cross-adsorbed https://www.thermofisher.com/antibody/product/Goat-anti-Rabbit-IgG-H-L-Highly-Cross-Adsorbed-Secondary-Antibody-Polyclonal/A-21245
rb A-21428 whole antibodies have been cross-adsorbed https://www.thermofisher.com/antibody/product/Goat-anti-Rabbit-IgG-H-L-Cross-Adsorbed-Secondary-Antibody-Polyclonal/A-21428

## Eukaryotic cell lines

Policy information about cell lines and Sex and Gender in Research

| | |
|---|---|
| Cell line source(s) | 168FARN and 67NR murine mammary tumor cell lines were obtained from the Karmanos Cancer Institute, HEK293T cells were obtained from ATCC and cultured following the provider's recommendations |
| Authentication | 168FARN and 67NR murine mammary tumor cell lines were directly obtained from the Karmanos Cancer Institute, HEK293T cells were directly obtained from ATCC. Thus, no further authentication has been done. |
| Mycoplasma contamination | Cell were frequently tested negative for mycoplasma using PCR-based assays. |
| Commonly misidentified lines (See ICLAC register) | No commonly misidentified cell line was used. |

## Animals and other research organisms

Policy information about studies involving animals; ARRIVE guidelines recommended for reporting animal research, and Sex and Gender in Research

| | |
|---|---|
| Laboratory animals | For mammary fatpad injection and intracardiac seeding female BALB/c or NOD.Cg-Prkdcscid Il2rgtm1Wjl/SzJ (NSG) mice were purchased from The Jackson Laboratory at 5-6 weeks of age. |
| Wild animals | The study did not involve wild animals. |
| Reporting on sex | All experiments were done in female mice only. Breast cancer rarely occurs in males thus research with female animals can be justified. |
| Field-collected samples | The study did not involve field-collected samples. |
| Ethics oversight | Dana-Farber Cancer Institute IACUC |

Note that full information on the approval of the study protocol must also be provided in the manuscript.

## Plants

| | |
|---|---|
| Seed stocks | Report on the source of all seed stocks or other plant material used. If applicable, state the seed stock centre and catalogue number. If plant specimens were collected from the field, describe the collection location, date and sampling procedures. |
| Novel plant genotypes | Describe the methods by which all novel plant genotypes were produced. This includes those generated by transgenic approaches, gene editing, chemical/radiation-based mutagenesis and hybridization. For transgenic lines, describe the transformation method, the number of independent lines analyzed and the generation upon which experiments were performed. For gene-edited lines, describe the editor used, the endogenous sequence targeted for editing, the targeting guide RNA sequence (if applicable) and how the editor was applied. |
| Authentication | Describe any authentication procedures for each seed stock used or novel genotype generated. Describe any experiments used to assess the effect of a mutation and, where applicable, how potential secondary effects (e.g. second site T-DNA insertions, mosiaicism, off-target gene editing) were examined. |

# ChIP-seq

## Data deposition

☒ Confirm that both raw and final processed data have been deposited in a public database such as GEO.

☒ Confirm that you have deposited or provided access to graph files (e.g. BED files) for the called peaks.

Data access links
*May remain private before publication.*

The following secure token has been created to allow review of record GSE237392 while it remains in private status: snwbeyicbtqffot

Files in database submission

20221207_MS11_MS10569_S163_R1_001.fastq.gz
20221207_MS11_MS10569_S163_R2_001.fastq.gz
20221207_MS20_MS10569_S29_R1_001.fastq.gz
20221207_MS20_MS10569_S29_R2_001.fastq.gz
20221207_MS21_MS10569_S30_R1_001.fastq.gz
20221207_MS21_MS10569_S30_R2_001.fastq.gz
20221207_MS26_MS10569_S34_R1_001.fastq.gz
20221207_MS26_MS10569_S34_R2_001.fastq.gz
20221207_MS39_MS10569_S65_R1_001.fastq.gz
20221207_MS39_MS10569_S65_R2_001.fastq.gz
20221207_MS40_MS10569_S66_R1_001.fastq.gz
20221207_MS40_MS10569_S66_R2_001.fastq.gz
20221207_MS51_MS10569_S76_R1_001.fastq.gz
20221207_MS51_MS10569_S76_R2_001.fastq.gz
20221207_MS52_MS10569_S77_R1_001.fastq.gz
20221207_MS52_MS10569_S77_R2_001.fastq.gz
20221207_MS53_MS10569_S78_R1_001.fastq.gz
20221207_MS53_MS10569_S78_R2_001.fastq.gz
20221207_MS54_MS10569_S79_R1_001.fastq.gz
20221207_MS54_MS10569_S79_R2_001.fastq.gz
20221207_MS55_MS10569_S80_R1_001.fastq.gz
20221207_MS55_MS10569_S80_R2_001.fastq.gz
20221208_MS50_MS10569R_S43_R1_001.fastq.gz
20221208_MS50_MS10569R_S43_R2_001.fastq.gz
220517_MS1_ACAGTG_MS10024_S5_R1_001.fastq.gz
220517_MS1_ACAGTG_MS10024_S5_R2_001.fastq.gz
220517_MS1_ACTTGA_MS10024_S8_R1_001.fastq.gz
220517_MS1_ACTTGA_MS10024_S8_R2_001.fastq.gz
220517_MS1_AGTCAA_MS10024_S13_R1_001.fastq.gz
220517_MS1_AGTCAA_MS10024_S13_R2_001.fastq.gz
220517_MS1_AGTTCC_MS10024_S14_R1_001.fastq.gz
220517_MS1_AGTTCC_MS10024_S14_R2_001.fastq.gz
220517_MS1_ATCACG_MS10024_S1_R1_001.fastq.gz
220517_MS1_ATCACG_MS10024_S1_R2_001.fastq.gz
220517_MS1_ATGTCA_MS10024_S15_R1_001.fastq.gz
220517_MS1_ATGTCA_MS10024_S15_R2_001.fastq.gz
220517_MS1_CAGATC_MS10024_S7_R1_001.fastq.gz
220517_MS1_CAGATC_MS10024_S7_R2_001.fastq.gz
220517_MS1_CCGTCC_MS10024_S16_R1_001.fastq.gz
220517_MS1_CCGTCC_MS10024_S16_R2_001.fastq.gz
220517_MS1_CGATGT_MS10024_S2_R1_001.fastq.gz
220517_MS1_CGATGT_MS10024_S2_R2_001.fastq.gz
220517_MS1_CTTGTA_MS10024_S12_R1_001.fastq.gz
220517_MS1_CTTGTA_MS10024_S12_R2_001.fastq.gz
220517_MS1_GATCAG_MS10024_S9_R1_001.fastq.gz
220517_MS1_GATCAG_MS10024_S9_R2_001.fastq.gz
220517_MS1_GCCAAT_MS10024_S6_R1_001.fastq.gz
220517_MS1_GCCAAT_MS10024_S6_R2_001.fastq.gz
220517_MS1_GGCTAC_MS10024_S11_R1_001.fastq.gz
220517_MS1_GGCTAC_MS10024_S11_R2_001.fastq.gz
220517_MS1_GTCCGC_MS10024_S17_R1_001.fastq.gz
220517_MS1_GTCCGC_MS10024_S17_R2_001.fastq.gz
220517_MS1_GTGAAA_MS10024_S18_R1_001.fastq.gz
220517_MS1_GTGAAA_MS10024_S18_R2_001.fastq.gz
220517_MS1_TAGCTT_MS10024_S10_R1_001.fastq.gz
220517_MS1_TAGCTT_MS10024_S10_R2_001.fastq.gz
220517_MS1_TGACCA_MS10024_S4_R1_001.fastq.gz
220517_MS1_TGACCA_MS10024_S4_R2_001.fastq.gz
220517_MS1_TTAGGC_MS10024_S3_R1_001.fastq.gz
220517_MS1_TTAGGC_MS10024_S3_R2_001.fastq.gz
220517_MS2_ACTTGA_MS10024_S26_R1_001.fastq.gz
220517_MS2_ACTTGA_MS10024_S26_R2_001.fastq.gz
220517_MS2_AGTCAA_MS10024_S31_R1_001.fastq.gz

```
220517_MS2_AGTCAA_MS10024_S31_R2_001.fastq.gz
220517_MS2_AGTTCC_MS10024_S32_R1_001.fastq.gz
220517_MS2_AGTTCC_MS10024_S32_R2_001.fastq.gz
220517_MS2_ATCACG_MS10024_S19_R1_001.fastq.gz
220517_MS2_ATCACG_MS10024_S19_R2_001.fastq.gz
220517_MS2_ATGTCA_MS10024_S33_R1_001.fastq.gz
220517_MS2_ATGTCA_MS10024_S33_R2_001.fastq.gz
220517_MS2_CCGTCC_MS10024_S34_R1_001.fastq.gz
220517_MS2_CCGTCC_MS10024_S34_R2_001.fastq.gz
220517_MS2_CTTGTA_MS10024_S30_R1_001.fastq.gz
220517_MS2_CTTGTA_MS10024_S30_R2_001.fastq.gz
220517_MS2_GATCAG_MS10024_S27_R1_001.fastq.gz
220517_MS2_GATCAG_MS10024_S27_R2_001.fastq.gz
220517_MS2_GCCAAT_MS10024_S24_R1_001.fastq.gz
220517_MS2_GCCAAT_MS10024_S24_R2_001.fastq.gz
220517_MS2_GGCTAC_MS10024_S29_R1_001.fastq.gz
220517_MS2_GGCTAC_MS10024_S29_R2_001.fastq.gz
220517_MS2_GTCCGC_MS10024_S35_R1_001.fastq.gz
220517_MS2_GTCCGC_MS10024_S35_R2_001.fastq.gz
220517_MS2_GTGAAA_MS10024_S36_R1_001.fastq.gz
220517_MS2_GTGAAA_MS10024_S36_R2_001.fastq.gz
220517_MS2_TAGCTT_MS10024_S28_R1_001.fastq.gz
220517_MS2_TAGCTT_MS10024_S28_R2_001.fastq.gz
220517_MS2_TGACCA_MS10024_S22_R1_001.fastq.gz
220517_MS2_TGACCA_MS10024_S22_R2_001.fastq.gz
220517_MS2_TTAGGC_MS10024_S21_R1_001.fastq.gz
220517_MS2_TTAGGC_MS10024_S21_R2_001.fastq.gz
122623_MSe01_MS11732_S33_R1_001.fastq.gz
122623_MSe01_MS11732_S33_R2_001.fastq.gz
122623_MSe02_MS11732_S34_R1_001.fastq.gz
122623_MSe02_MS11732_S34_R2_001.fastq.gz
122623_MSe03_MS11732_S35_R1_001.fastq.gz
122623_MSe03_MS11732_S35_R2_001.fastq.gz
122623_MSe06_MS11732_S38_R1_001.fastq.gz
122623_MSe06_MS11732_S38_R2_001.fastq.gz
122623_MSe07_MS11732_S39_R1_001.fastq.gz
122623_MSe07_MS11732_S39_R2_001.fastq.gz
122623_MSe08_MS11732_S40_R1_001.fastq.gz
122623_MSe08_MS11732_S40_R2_001.fastq.gz
122623_MSe10_MS11732_S42_R1_001.fastq.gz
122623_MSe10_MS11732_S42_R2_001.fastq.gz
122623_MSe11_MS11732_S43_R1_001.fastq.gz
122623_MSe11_MS11732_S43_R2_001.fastq.gz
122623_MSe12_MS11732_S44_R1_001.fastq.gz
122623_MSe12_MS11732_S44_R2_001.fastq.gz
220517_MS1_AGTCAA_MS10024_S13_R1_001.bw
220517_MS1_AGTTCC_MS10024_S14_R1_001.bw
220517_MS1_ATGTCA_MS10024_S15_R1_001.bw
220517_MS1_CCGTCC_MS10024_S16_R1_001.bw
220517_MS1_CTTGTA_MS10024_S12_R1_001.bw
220517_MS1_GGCTAC_MS10024_S11_R1_001.bw
220517_MS1_GTCCGC_MS10024_S17_R1_001.bw
220517_MS1_GTGAAA_MS10024_S18_R1_001.bw
220517_MS1_TAGCTT_MS10024_S10_R1_001.bw
220517_MS2_AGTCAA_MS10024_S31_R1_001.bw
220517_MS2_AGTTCC_MS10024_S32_R1_001.bw
220517_MS2_ATGTCA_MS10024_S33_R1_001.bw
220517_MS2_CCGTCC_MS10024_S34_R1_001.bw
220517_MS2_CTTGTA_MS10024_S30_R1_001.bw
220517_MS2_GGCTAC_MS10024_S29_R1_001.bw
220517_MS2_GTCCGC_MS10024_S35_R1_001.bw
220517_MS2_GTGAAA_MS10024_S36_R1_001.bw
220517_MS2_TAGCTT_MS10024_S28_R1_001.bw

168_KMT2C_Mut_H3K27me3_broad_a.bw
168_KMT2C_Mut_H3K27me3_broad_c.bw
168_KMT2C_Mut_KDM6A_broad_a.bw
168_KMT2C_Mut_KDM6A_broad_b.bw
168_KMT2C_MUT_P300_a.bw
168_KMT2C_MUT_P300_b.bw
168_KMT2D_Mut_H3K27me3_broad_a.bw
168_KMT2D_Mut_H3K27me3_broad_c.bw
168_KMT2D_Mut_KDM6A_broad_a.bw
168_KMT2D_Mut_KDM6A_broad_b.bw
168_KMT2D_MUT_P300_a.bw
168_KMT2D_MUT_P300_b.bw
```

168_WT_H3K27me3_broad_a.bw
168_WT_H3K27me3_broad_c.bw
168_WT_KDM6A_broad_a.bw
168_WT_KDM6A_broad_b.bw
168_WT_P300_a.bw
168_WT_P300_b.bw

**Genome browser session**
(e.g. UCSC)

no longer applicable

## Methodology

**Replicates**

For each ChIPseq 2 or 3 biological replicates (defined as 1 passaging (~5 days) between collection of cell line replicates) were sequenced.

**Sequencing depth**

Libraries were sequenced on an Illumina NovaSeq6000 (Illumina) targeting 40 million 150bp read pairs by the Molecular Biology Core facilities at Dana-Farber Cancer Institute.

**Antibodies**

H3K4me1 rb 2.5 ug/ChIP abcam ab8895 ChIP
H3K27me3 rb 2.5 ug/ChIP Cell Signaling 9733S ChIP
H3K27ac rb 2.5 ug/ChIP Diagenode C15410196 ChIP
KDM6A rb 2.5 ug/ChIP Cell Signaling 33510S ChIP
P300 rb 2.5 ug/ChIP abcam ab275378 ChIP

**Peak calling parameters**

All data were analyzed using mm9 genome. Mapping was done using BWA-MEM and peaks were called using macs2 (callpeak --SPMR --broad -B -q 0.01 --keep-dup 1 -g 1.87e9 -f BAMPE --extsize 146 --nomodel). --broad was excluded for H3K27ac peak calling.

**Data quality**

Minimum number of peaks with >10FC enrichment at 5% FDR was 400 for any dataset. Random genome tracks in each dataset have been visually analyzed for peak shape. Further quality control was performed with ChIPQC. Minimum percentage of reads passing mapping QC filter was 80. No reads were reported as duplicates.

**Software**

Analysis was performed using the COBRA pipeline (https://bitbucket.org/cfce/cobra/src/master/). For differential peak analysis adj.p.value < 0.05 and log2FC >0.5 were used.novo motifs were considered. For motif heatmap only motifs with -log10 p-value > 25 in at least one sample were plotted. Correlation of differential peaks and gene expression was performed using BETA18. Each individual differential gained or lost peakset was analyzed together with DESeq2 output from RNA-seq of the same cells but different biological replicates using default parameters. For quantification of P300 signal intensities within H3K27ac peaks multiBigwigSummary was used to bin P300 signal of each biological replicate into specific peak regions. Bin count of replicates was averaged and log2 transformed. Density plots for correlation of changed P300 and H3K27ac signal was generated with ggplot. DEseq2 outputs from CoBRA pipeline were used to extract all calculated fold changes within identified peak regions for H3K27ac or P300 in Kmt2c or Kmt2d KO cells compared to WT. Then multiBigwigSummary43 was used with a 100 bin size to calculate fold changes of H3K27ac or P300 in similar region. Quantification of read counts for H3K27me3, P300 and KDM6A within the Mmp3 promoter (TSS +/- 1kb) was performed using featureCounts44 with respective bam files. For comparisons of signal intensities across the Mmp3 cluster locus signal of biological replictaes was first averaged using bigwigAverage43. Then average values were compared using bigwigCompare43 in 1000 bp bin and log2FC mode. Overall signal intensity was then calculated from average values of all bins.

# Flow Cytometry

## Plots

Confirm that:

☒ The axis labels state the marker and fluorochrome used (e.g. CD4-FITC).

☒ The axis scales are clearly visible. Include numbers along axes only for bottom left plot of group (a 'group' is an analysis of identical markers).

☒ All plots are contour plots with outliers or pseudocolor plots.

☒ A numerical value for number of cells or percentage (with statistics) is provided.

## Methodology

**Sample preparation**

Immediately after collection tissues were smashed with micro pestles and digested for 10 min (brain, bone marrow) or 1 h (tumor, liver and lung) using digestion media (2% w/v collagenase IV, 2% w/v hyaluronidase and 2% w/v BSA in DMEM) at 37 C on a shaker. Solutions were filtered through a mesh, washed with PBS and frozen in 10% DMSO/FBS at -80 C or directly used for flow cytometry. For this, cells were passed through 70 or 100 m cell strainer, incubated with DAPI (1:20,000) and analyzed using a LSR Fortessa (BD Biosciences). Gating strategies can be found in Supplementary Fig 1

**Instrument**

BD LSRFortessa™ Cell Analyzer

**Software**

BD FACSDiva was used for data acquisition and FlowJo version 10.9.0 was used for data analysis.

**Cell population abundance**

No sorting has been performed. Abundance was the primary measurement and thus can be found in the according source data and figures. Purity was controlled with DAPI exclusion.

**Gating strategy**

Gating strategies can be found in Supplementary Fig 1. In short, cells were selected using FSC-A/SSC-A, single cells were

Gating strategy | selected using SSC-H/SSC-W or FSC-A/FSC-H. DAPI, GFP and mCherry positive or negative cells were selected using unstained negative controls and stained positive controls.

☒ Tick this box to confirm that a figure exemplifying the gating strategy is provided in the Supplementary Information.

