## [Peer Review File · Nature Cell Biology]

Peer Review Information

Journal: Nature Cell Biology

Manuscript Title: Loss of Kmt2c or Kmt2d drives brain metastasis via KDM6A-dependent upregulation of MMP3

Corresponding author name(s): Kornelia Polyak

Editorial Notes:

Reviewer Comments & Decisions:

Decision Letter, initial version:

Dear Dr Polyak,

Please first accept our apology for the delay getting back to you with a decision due to difficulties in retrieving reviewer's comments.

Your manuscript, "Loss of Kmt2c or Kmt2d drives brain metastasis via KDM6A-dependent upregulation of MMP3", has now been seen by 2 referees, who are experts in breast cancer, metastasis (referee 2); and breast cancer, immune microenvironment (referee 3). As you will see from their comments (attached below) they find this work of potential interest, but have raised substantial concerns, which in our view would need to be addressed with considerable revisions before we can consider publication in Nature Cell Biology.

We would like to clarify that although we have engaged a third referee (Reviewer 1) with expertise on cancer epigenetics on the referee panel, the comments raised by the other two referees were sufficient for us to form a decision in the absence of this expert's feedback, and we felt a further delay would be counterproductive for the authors. We will send you the third report if/when we receive it.

Nature Cell Biology editors discuss the referee reports in detail within the editorial team, including the chief editor, to identify key referee points that should be addressed with priority, and requests that are overruled as being beyond the scope of the current study. To guide the scope of the revisions, I have listed these points below. We are committed to providing a fair and constructive peer-review process, so please feel free to contact me if you would like to discuss any of the referee comments further.

In particular, it would be essential to:

A) Strengthen the proposed mechanism for KMT2 and KDM6A:

Reviewer 1

"Given the potential importance of the COMPASS complex's formation in the regulation of MMP3 without catalytic functions of KMT2, it becomes essential to delve into the specific mechanisms by which KDM6A governs MMP3 and whether MMP3 is a direct target of KDM6A. The author should examine the H3K27me3 levels or KDM6A binding at the MMP3 locus in both KMT2 KO cells to validate this point."

"The KMT2 complex recruits p300/CBP (H3K27 acetyltransferase) through KDM6A to initiate and sustain the deposition of H3K4me1 and H3K27ac at enhancer regions (Biochem Soc Trans. 2021 Jun 30; 49(3): 1041–1054.). Notably, the data reveals an elevation in H3K27ac levels at the MMP3 locus within KMT2 KO cells. Furthermore, the impact of GSK-J4 treatment on H3K27me3 levels is distinct from its effect on H3K27ac: while it maintains the global H3K27me3 level, it reduces H3K27ac levels even though GSK-J4 is well-known to increase global H3K27me3 as a KDM6A inhibitor. This intriguing observation hints at the potential critical role of recruiting P300/CBP to the MMP3 locus by KDM6A, suggesting a mechanism through which MMP3 upregulation might occur. It is recommended that the authors explore and elucidate this intriguing phenomenon in their analysis."

B) Clarify the inconsistency with the literature:

Reviewer 1

"KMT2 is widely recognized for its role in catalyzing H3K4 mono-methylation as a methyltransferase. Depletion of KMT2 has been observed to lead to a decrease in H3K4me1 levels, accompanied by a reduction in H3K27ac at both promoter and enhancer regions (Oncogene. 2018; 37(34): 4692–4710., Cell Rep. 2021 Feb 16; 34(7): 108751., Cell Death Dis. 2021 Apr; 12(4): 364.). However, this study presents an intriguing finding where the loss of KMT2 results in an elevated H3K4me1 level specifically at the MMP3 locus. This observation suggests that KMT2 might govern the KDM6A-MMP3 through a mechanism independent of its canonical methyltransferase activity or via an indirect pathway. The authors should address this point."

C) All other referee concerns pertaining to strengthening existing data, providing controls, methodological details, clarifications and textual changes as applicable should also be addressed.

D) Finally please pay close attention to our guidelines on statistical and methodological reporting (listed below) as failure to do so may delay the reconsideration of the revised manuscript. In particular please provide:

We would be happy to consider a revised manuscript that would satisfactorily address these points, unless a similar paper is published elsewhere, or is accepted for publication in Nature Cell Biology in the meantime.

- ensure that it conforms to our format instructions and publication policies (see below and <https://www.nature.com/nature/for-authors>).
- provide a point-by-point rebuttal to the full referee reports verbatim, as provided at the end of this letter.
- provide the completed Reporting Summary (found here <https://www.nature.com/documents/nr-reporting-summary.pdf>). This is essential for reconsideration of the manuscript will be available to editors and referees in the event of peer review. For more information see <http://www.nature.com/authors/policies/availability.html> or contact me.

When submitting the revised version of your manuscript, please pay close attention to our [href="https://www.nature.com/nature-portfolio/editorial-policies/image-integrity">Digital Image Integrity Guidelines](https://www.nature.com/nature-portfolio/editorial-policies/image-integrity). and to the following points below:

Nature Cell Biology is committed to improving transparency in authorship. As part of our efforts in this direction, we are now requesting that all authors identified as 'corresponding author' on published papers create and link their Open Researcher and Contributor Identifier (ORCID) with their account on the Manuscript Tracking System (MTS), prior to acceptance. ORCID helps the scientific community achieve unambiguous attribution of all scholarly contributions. You can create and link your ORCID from the home page of the MTS by clicking on 'Modify my Springer Nature account'. For more information please visit www.springernature.com/orcid.

This journal strongly supports public availability of data. Please place the data used in your paper into a public data repository, or alternatively, present the data as Supplementary Information. If data can only be shared on request, please explain why in your Data Availability Statement, and also in the correspondence with your editor. Please note that for some data types, deposition in a public repository is mandatory - more information on our data deposition policies and available repositories appears below.

[REDACTED]

We would like to receive a revised submission within six months.

We hope that you will find our referees' comments, and editorial guidance helpful. Please do not hesitate to contact me if there is anything you would like to discuss.

Best wishes,

Zhe Wang

Zhe Wang, PhD
Senior Editor
Nature Cell Biology

Tel: +44 (0) 207 843 4924
email: zhe.wang@nature.com

Reviewers' Comments:

Reviewer #2:

Remarks to the Author:

This manuscript reported the effects of KMT2C or KMT2D deletion on breast cancer brain metastasis. Specifically, the authors elucidated the impact of KMT2C or KMT2D deletion in TNBC, revealing their propensity to induce brain metastasis by reshaping cancer cell histone profiles regardless of a functional immune system. The alteration of H3K4me1, H3K27ac, and H3K27me3 is uncovered, which is linked to increased binding of the KDM6A enzyme. Further, the data indicated an upregulation of the Mmp3 gene is observed in both models, with clinical validation in TNBC cases. Notably, the inhibition of KDM6A mitigates the KMT2-loss-induced upregulation of Mmp3 and consequently prevents brain metastasis, suggesting the KDM6A-MMP3 axis as a pivotal driver of KMT2-loss-induced metastasis. The manuscript is clearly written with well-designed experiments, but the authors should address the following points during the revision of the manuscript.

1. KMT2 is widely recognized for its role in catalyzing H3K4 mono-methylation as a methyltransferase. Depletion of KMT2 has been observed to lead to a decrease in H3K4me1 levels, accompanied by a reduction in H3K27ac at both promoter and enhancer regions (Oncogene. 2018; 37(34): 4692–4710., Cell Rep. 2021 Feb 16; 34(7): 108751., Cell Death Dis. 2021 Apr; 12(4): 364.). However, this study presents an intriguing finding where the loss of KMT2 results in an elevated H3K4me1 level specifically

at the MMP3 locus. This observation suggests that KMT2 might govern the KDM6A-MMP3 through a mechanism independent of its canonical methyltransferase activity or via an indirect pathway. The authors should address this point.

2. Given the potential importance of the COMPASS complex's formation in the regulation of MMP3 without catalytic functions of KMT2, it becomes essential to delve into the specific mechanisms by which KDM6A governs MMP3 and whether MMP3 is a direct target of KDM6A. The author should examine the H3K27me3 levels or KDM6A binding at the MMP3 locus in both KMT2 KO cells to validate this point.

3. The KMT2 complex recruits p300/CBP (H3K27 acetyltransferase) through KDM6A to initiate and sustain the deposition of H3K4me1 and H3K27ac at enhancer regions (Biochem Soc Trans. 2021 Jun 30; 49(3): 1041–1054.). Notably, the data reveals an elevation in H3K27ac levels at the MMP3 locus within KMT2 KO cells. Furthermore, the impact of GSK-J4 treatment on H3K27me3 levels is distinct from its effect on H3K27ac: while it maintains the global H3K27me3 level, it reduces H3K27ac levels even though GSK-J4 is well-known to increase global H3K27me3 as a KDM6A inhibitor. This intriguing observation hints at the potential critical role of recruiting P300/CBP to the MMP3 locus by KDM6A, suggesting a mechanism through which MMP3 upregulation might occur. It is recommended that the authors explore and elucidate this intriguing phenomenon in their analysis.

4. The authors have illustrated the prophylactic effect of GSK-J4 in Figure 4. However, a comprehensive assessment of the in vivo efficacy of GSK-J4 would greatly enhance the understanding of its therapeutic potential. Further analysis and characterization of the treatment's impact on relevant biological parameters, such as metastatic burden, biomarker expression, and overall survival rates, could provide valuable insights into the true effectiveness of GSK-J4 as a therapeutic intervention. This additional evaluation would offer a more robust basis for drawing conclusions about the potential clinical application of GSK-J4 to prevent metastasis of KMT2C or KMT2D mutant tumors.

5. On page 10, line 9, the authors mentioned Figure 3o, but there is no Figure 3o.

Reviewer #3:

Remarks to the Author:

Well prepared article which addresses how KMT2C and 2D mutations shape epigenomics and transcriptomic landscapes to promote metastasis.

Have only a minor concern with methodology used. Why have you used FACS for metastasis assessment rather than qPCR for mcherry which is standard. Has this been validated? Also for IF staining was it a pressure cooker and if so what settings? Do you mean goat anti-rabbit (page 23)?

Other queries:

1. How prevalent are MMP3 high TNBC tumours with KMT2C mutations?
2. Page 4 Immunoblot analysis confirmed reduced expression in the KO lines - can you mention how much it was reduced.
3. gsk-j4 sensitivity was not changed much 0.45nM vs ~0.2 - was this expected?
4. Why couldn't you use a more physiological metastasis assay (from MFP)?
5. Figure 1 - why is it brain specific? Can you align this with published work looking at site specific mets.
6. Figure 4d shows MMP3 is higher in TNBC with KMT2C mutations - what about in mets versus primary?
7. Figure 4F.G - Were mets seen in any other locations?

8. Why was only 2 days of GSK-J4 treatment used?
9. what is the likelihood/path for translation?
10. Figure 2o - the images are very hard to see and thus not convincing
11. Can statistical analyses be performed on the immune changes assessed in Fig 2

The abstract and MS are well written, but it could be improved if you could explain how this would be translated.

Methods should be written concisely, but should contain all elements necessary to allow interpretation and replication of the results. As a guideline, Methods sections typically do not exceed 3,000 words. The Methods should be divided into subsections listing reagents and techniques. When citing previous methods, accurate references should be provided and any alterations should be noted. Information must be provided about: antibody dilutions, company names, catalogue numbers and clone numbers for monoclonal antibodies; sequences of RNAi and cDNA probes/primers or company names and catalogue numbers if reagents are commercial; cell line names, sources and information on cell line identity and authentication. Animal studies and experiments involving human subjects must be reported in detail, identifying the committees approving the protocols. For studies involving human subjects/samples, a statement must be included confirming that informed consent was obtained. Statistical analyses and information on the reproducibility of experimental results should be provided in a section titled "Statistics and Reproducibility".

All Nature Cell Biology manuscripts submitted on or after March 21 2016 must include a Data availability statement as a separate section after Methods but before references, under the heading "Data Availability". For Springer Nature policies on data availability see <http://www.nature.com/authors/policies/availability.html>; for more information on this particular policy see <http://www.nature.com/authors/policies/data/data-availability-statements-data-citations.pdf>. The Data availability statement should include:

- Accession codes for primary datasets (generated during the study under consideration and

designated as "primary accessions") and secondary datasets (published datasets reanalysed during the study under consideration, designated as "referenced accessions"). For primary accessions data should be made public to coincide with publication of the manuscript. A list of data types for which submission to community-endorsed public repositories is mandated (including sequence, structure, microarray, deep sequencing data) can be found here <http://www.nature.com/authors/policies/availability.html#data>.

- Unique identifiers (accession codes, DOIs or other unique persistent identifier) and hyperlinks for datasets deposited in an approved repository, but for which data deposition is not mandated (see here for details <http://www.nature.com/sdata/data-policies/repositories>).
- At a minimum, please include a statement confirming that all relevant data are available from the authors, and/or are included with the manuscript (e.g. as source data or supplementary information), listing which data are included (e.g. by figure panels and data types) and mentioning any restrictions on availability.
- If a dataset has a Digital Object Identifier (DOI) as its unique identifier, we strongly encourage including this in the Reference list and citing the dataset in the Methods.

We recommend that you upload the step-by-step protocols used in this manuscript to the Protocol Exchange. More details can be found at www.nature.com/protocolexchange/about.

All imaging data should be accompanied by scale bars, which should be defined in the legend. Cropped images of gels/blots are acceptable, but need to be accompanied by size markers, and to retain visible background signal within the linear range (i.e. should not be saturated). The boundaries of panels with low background have to be demarked with black lines. Splicing of panels should only be considered if unavoidable, and must be clearly marked on the figure, and noted in the legend with a statement on whether the samples were obtained and processed simultaneously. Quantitative comparisons between samples on different gels/blots are discouraged; if this is unavoidable, it should only be performed for samples derived from the same experiment with gels/blots were processed in parallel, which needs to be stated in the legend.

Figures should be provided at approximately the size that they are to be printed at (single column is 86 mm, double column is 170 mm) and should not exceed an A4 page (8.5 x 11"). Reduction to the scale that will be used on the page is not necessary, but multi-panel figures should be sized so that the whole figure can be reduced by the same amount at the smallest size at which essential details in each panel are visible. In the interest of our colour-blind readers we ask that you avoid using red and green for contrast in figures. Replacing red with magenta and green with turquoise are two possible

colour-safe alternatives. Lines with widths of less than 1 point should be avoided. Sans serif typefaces, such as Helvetica (preferred) or Arial should be used. All text that forms part of a figure should be rewritable and removable.

SUPPLEMENTARY INFORMATION – Supplementary information is material directly relevant to the conclusion of a paper, but which cannot be included in the printed version in order to keep the manuscript concise and accessible to the general reader. Supplementary information is an integral

part of a Nature Cell Biology publication and should be prepared and presented with as much care as the main display item, but it must not include non-essential data or text, which may be removed at the editor's discretion. All supplementary material is fully peer-reviewed and published online as part of the HTML version of the manuscript. Supplementary Figures and Supplementary Notes are appended at the end of the main PDF of the published manuscript.

The total number of Supplementary Figures (not including the "unprocessed scans" Supplementary Figure) should not exceed the number of main display items (figures and/or tables (see our Guide to Authors and March 2012 editorial <http://www.nature.com/ncb/authors/submit/index.html#suppinfo>; <http://www.nature.com/ncb/journal/v14/n3/index.html#ed>). No restrictions apply to Supplementary Tables or Videos, but we advise authors to be selective in including supplemental data.

GUIDELINES FOR EXPERIMENTAL AND STATISTICAL REPORTING

REPORTING REQUIREMENTS – We are trying to improve the quality of methods and statistics reporting in our papers. To that end, we are now asking authors to complete a reporting summary that collects information on experimental design and reagents. The Reporting Summary can be found here <https://www.nature.com/documents/nr-reporting-summary.pdf> If you would like to reference the guidance text as you complete the template, please access these flattened versions at <http://www.nature.com/authors/policies/availability.html>.

STATISTICS – Wherever statistics have been derived the legend needs to provide the n number (i.e. the sample size used to derive statistics) as a precise value (not a range), and define what this value represents. Error bars need to be defined in the legends (e.g. SD, SEM) together with a measure of centre (e.g. mean, median). Box plots need to be defined in terms of minima, maxima, centre, and percentiles. Ranges are more appropriate than standard errors for small data sets. Wherever statistical significance has been derived, precise p values need to be provided and the statistical test used needs to be stated in the legend. Statistics such as error bars must not be derived from $n < 3$. For

sample sizes of $n < 5$ please plot the individual data points rather than providing bar graphs. Deriving statistics from technical replicate samples, rather than biological replicates is strongly discouraged. Wherever statistical significance has been derived, precise p values need to be provided and the statistical test stated in the legend.

Author Rebuttal to Initial comments

We thank the Reviewers for their critical reading of our manuscript, positive comments, and useful suggestions on how to improve it. We performed several additional experiments to address the reviewers' comments and revised the manuscript accordingly. Below are our point-by-point responses (in blue) to each of the reviewers' specific points.

Reviewer #2:

Remarks to the Author:

This manuscript reported the effects of KMT2C or KMT2D deletion on breast cancer brain metastasis. Specifically, the authors elucidated the impact of KMT2C or KMT2D deletion in TNBC, revealing their propensity to induce brain metastasis by reshaping cancer cell histone profiles regardless of a functional immune system. The alteration of H3K4me1, H3K27ac, and H3K27me3 is uncovered, which is linked to increased binding of the KDM6A enzyme. Further, the data indicated an upregulation of the Mmp3 gene is observed in both models, with clinical validation in TNBC cases. Notably, the inhibition of KDM6A mitigates the KMT2-loss-induced upregulation of Mmp3 and consequently prevents brain metastasis, suggesting the KDM6A-MMP3 axis as a pivotal driver of KMT2-loss-induced metastasis. The manuscript is clearly written with well-designed experiments, but the authors should address the following points during the revision of the manuscript.

We appreciate the reviewer's positive comments on our manuscript.

1. KMT2 is widely recognized for its role in catalyzing H3K4 mono-methylation as a methyltransferase.

Depletion of KMT2 has been observed to lead to a decrease in H3K4me1 levels, accompanied by a reduction in H3K27ac at both promoter and enhancer regions (Oncogene. 2018; 37(34): 4692–4710., Cell Rep. 2021 Feb 16; 34(7): 108751., Cell Death Dis. 2021 Apr; 12(4): 364.). However, this study presents an intriguing finding where the loss of KMT2 results in an elevated H3K4me1 level specifically at the MMP3 locus. This observation suggests that KMT2 might govern the KDM6A-MMP3 through a mechanism independent of its canonical methyltransferase activity or via an indirect pathway. The authors should address this point.

This an excellent point. To address it, we first tested if inhibition of KDM1A, the major H3K4 demethylase, affects *Mmp3* expression, but found no change in *Mmp3* mRNA in any of the cell lines even though H3K4me1 was increased (**new Fig. 4I and Extended Data Figure 6a**). These data suggest that H3K4me1 levels do not directly affect *Mmp3* expression in agreement with prior studies reporting that H3K27me3 and H3K4me1 inversely correlate depending on KDM6A activity (Duplaquet et al. Nat Cell Biol 2023) and that bimodal H3K4me1 promoter signal is associated with higher gene activity (Bae et al. Front Cell Dev Biol. 2020) and thus we might see an indirect increase in H3K4me1. We cited these publications and included a paragraph in the discussion:

*“This might also explain our finding of increased H3K4me1 and H3K27ac signal upon *Kmt2c* or *Kmt2d* loss as opposed to other publications. A recent study showed that H3K27me3 and H3K4me1 inversely correlate in a KDM6A-dependent manner (Duplaquet et al. Nat Cell Biol 2023), suggesting an indirect impact of KDM6A on H3K4me1 regulation. It has also been shown that higher bimodal H3K4me1 signal at promoters correlates with increased gene activity (Bae et al. Front Cell Dev Biol. 2020) further suggesting H3K4me1 is also indirectly regulated in our *Kmt2c* and *Kmt2d* KO cells.”*

We further tested if *Kmt2c* or *Kmt2d* KO leads to a disruption of KDM6A bound COMPASS complex, however, we did not see any difference of known COMPASS member bound to immunoprecipitated KDM6A (**new Extended Data Fig.6g**). Importantly, as described in the answers to the comments below, our new data suggest a deregulation of non-promoter activity in *Kmt2c* and *Kmt2d* KO cells through KDM6A leading to increased gene expression co-occurring with P300-mediated H3K27 acetylation at promoters. This suggest an indirect activation of the KDM6A-MMP3 axis in *Kmt2* KO cells via P300.

2. Given the potential importance of the COMPASS complex's formation in the regulation of

MMP3 without catalytic functions of KMT2, it becomes essential to delve into the specific mechanisms by which KDM6A governs MMP3 and whether MMP3 is a direct target of KDM6A. The author should examine the H3K27me3 levels or KDM6A binding at the MMP3 locus in both KMT2 KO cells to validate this point.

Thank you for this suggestion. To address this point, we analyzed H3K27me3, KDM6A, and newly generated P300 ChIP-seq data for signal within the *Mmp3* promoter. We detected lower H3K27me3 and higher P300 signal intensity in both *Kmt2c* and *Kmt2d* KO cells compared to WT at the *Mmp3* promoter, while there was no clear pattern for KDM6A (**new Ext data Fig. 6b**).

We further expanded this analysis to the whole *Mmp* gene cluster locus which has been reported to be coregulated in breast cancer (Llinas-Arias, P. et al. Mol Cancer 2023), and found similar results: gained H3K4me1, H3K27ac, and P300 signal, lower H3K27me3, and variable KDM6A binding (**new Extended Data Fig 6c,d**).

We also tested the effects of P300 and KDM6A inhibitors and found that they decreased the expression of *Mmp3* and other *Mmp cluster* genes in *Kmt2c/d* KO cells (**new Fig. 4k, I, Extended Data Figure 5f, g, 6a and f**).

Overall, these data suggest that P300 can directly influence *Mmp3* expression via enhancing H3K27ac at promoter regions while KDM6A might act at non-promoter sites. In line with this, correlation of gained KDM6A peaks with upregulated genes was lost when we only focused on peaks in promoter regions (**new Extended Data Fig. 6h**) but gained significance when we considered peaks with increasing distance from TSS (**new Extended Data Fig. 6i**).

3. The KMT2 complex recruits p300/CBP (H3K27 acetyltransferase) through KDM6A to initiate and sustain the deposition of H3K4me1 and H3K27ac at enhancer regions (Biochem Soc Trans. 2021 Jun 30; 49(3): 1041–1054.). Notably, the data reveals an elevation in H3K27ac levels at the MMP3 locus within KMT2 KO cells. Furthermore, the impact of GSK-J4 treatment on H3K27me3 levels is distinct from its effect on H3K27ac: while it maintains the global H3K27me3 level, it reduces H3K27ac levels even though GSK-J4 is well-known to increase global H3K27me3 as a KDM6A inhibitor. This intriguing observation hints at the potential critical role of recruiting P300/CBP to the MMP3 locus by KDM6A, suggesting a mechanism through which MMP3 upregulation might occur. It is recommended that the authors explore and elucidate this intriguing phenomenon in their analysis.

Thank you for raising this important point. We performed P300 ChIP-seq in our WT and *Kmt2c/d* KO cells. Although we only found a few significantly differential peaks between KO and WT cells, P300 signal intensity was significantly higher in regions with gained H3K27ac peaks in both KO cells (**new Fig. 3h**). Furthermore, global differential peak intensities of P300 significantly

correlated with differential peak intensities of H3K27ac in both KO cells compared to WT (**new Extended Data Figure 3b**). Together with the data described above, this suggests that the increase of H3K27ac is the result of P300 activity with a simultaneous increase in gene activation correlating with KDM6A binding at non-promoter sites.

4. The authors have illustrated the prophylactic effect of GSK-J4 in Figure 4. However, a comprehensive assessment of the in vivo efficacy of GSK-J4 would greatly enhance the understanding of its therapeutic potential. Further analysis and characterization of the treatment's impact on relevant biological parameters, such as metastatic burden, biomarker expression, and overall survival rates, could provide valuable insights into the true effectiveness of GSK-J4 as a therapeutic intervention. This additional evaluation would offer a more robust basis for drawing conclusions about the potential clinical application of GSK-J4 to prevent metastasis of KMT2C or KMT2D mutant tumors.

Following the reviewer's suggestion, we performed an in vivo treatment study with GSK-J4. We pre-treated both the cells and mice with GSK-J4 for 2 days prior to intracardiac injection, and continued treatment until day 12, when all mice were euthanized. Similar to our prior observation (**now Extended Data Fig. 7 e,f**) we saw a significant decrease of metastases in *Kmt2c* KO group and a trend for decreased metastatic burden in the *Kmt2d* KO group upon GSK-J4 treatment (**new Fig. 4m and Extended Data Figure 7a,c**) but no change in bone metastases (**new Extended Data Figure 7d**), and no histologic changes in any major organs (**Extended Data Figure 7b**). These data suggest that inhibition of KDM6A is a potential therapeutic strategy for the prevention of brain metastases in patient with *KMT2C/KMT2D* mutant triple-negative breast cancer.

5. On page 10, line 9, the authors mentioned Figure 3o, but there is no Figure 3o.

Thank you for pointing this out. We corrected this mistake.

Reviewer #3:

Remarks to the Author:

Well prepared article which addresses how KMT2C and 2D mutations shape epigenomics and transcriptomic landscapes to promote metastasis.

We appreciate the reviewer's positive comments on our manuscript.

Have only a minor concern with methodology used. Why have you used FACS for metastasis assessment rather than qPCR for mCherry which is standard. Has this been validated?

The rationale for using FACS was that we can specifically quantify viable metastatic cells and avoid potential artifacts due to differences in mCherry expression among cells driven by differences in genomic integration site. However, to directly address the reviewer's comment, we performed qPCR for mCherry and correlated this with the ratio of mCherry positive cells measured via FACS from the same samples. We found a significant correlation between the two different assays (**new Extended data Fig 1o**) and the conclusions remained the same.

Also for IF staining was it a pressure cooker and if so what settings? Do you mean goat anti-rabbit (page 23)?

For antigen retrieval prior to IF staining we used incubation in a food steamer for 40 min, we did not use a pressure cooker.

That is correct, anti rb is an abbreviation for anti-rabbit. We included these in the Methods section of the revised manuscript.

Other queries:

1. How prevalent are MMP3 high TNBC tumours with KMT2C mutations?

To investigate this, we analyzed the METABRIC TNBC cohort and divided patients into *MMP3* low (bottom 50%) and *MMP3* high (top 50%) groups based on *MMP3* mRNA levels and analyzed the frequency of *KMT2C* and *KMT2D* mutant tumors in each group. Similar to our prior results in Fig. 4e, we found that the proportion of *KMT2C* mutant tumors is significantly higher in the *MMP3* high compared to *MMP3* low TNBC group, while *KMT2D* mutation frequencies did not show any difference (**new Extended Data Figure 5a**).

2. Page 4 Immunoblot analysis confirmed reduced expression in the KO lines - can you mention how much it was reduced.

We performed densitometric analyses on the western blot images and added the values to the images in Fig. 1c and Extended Data Figure 1f. All uncropped images are included in Supplementary Figure 2.

3. gsk-j4 sensitivity was not changed much 0.45nM vs ~0.2 - was this expected?

Our data suggests that reduced expression of *Kmt2c* or *Kmt2d* increases KDM6A binding and potentially its activity as well. However, this does not necessarily lead to a significant increase in KDM6A IC50. To explore this in more detail, we investigated sensitivity to GSK-J4 in 25 different human breast cancer cells in the DepMap database and correlated this with *KMT2C*, *KMT2D*, and KDM6A expression level (**rebuttal Figure 1**). We did not observe a significant correlation between the expression of any of these genes and GSK-J4 sensitivity providing additional evidence that lower levels of *Kmt2c* or *Kmt2d* do not significantly impact sensitivity to gsk-j4.

Rebuttal Figure 1: Correlation of GSK-J4 sensitivity and *KMT2C* (left) or *KMT2D* (right) expression level from DepMap database.

4. Why couldn't you use a more physiological metastasis assay (from MFP)?

Thank you for raising this point. Our first observation of increased metastases in the *Kmt2c/Kmt2d* KO models compared to WT was based on mammary fat pad injections (**Fig. 1f**). However, the frequency of metastases was too low for reliable quantification and statistical analyses using reasonable numbers of mice. Thus, we performed intracardiac injections, a widely accepted model for experimental metastases, in subsequent experiments.

5. Figure 1 - why is it brain specific? Can you align this with published work looking at site specific mets.

We apologize for implying that *Kmt2c* or *Kmt2d* mutations are only associated with brain metastases. In the matched primary TNBC/distant metastases cohort we used in Figure 1, we also found significantly lower *KMT2C* expression in liver metastases (**rebuttal Figure 2**) implying loss of *KMT2* might also drive metastases to other organs. Using our cell line model, we were not able to reliably detect liver metastases, however, we also found increased bone metastases in the KO cell lines (**Extended Data Figure 1p and q**). Thus, loss of *KMT2* can lead to multi-organ metastases, but we focused on brain metastases in this manuscript as this was the most robust phenotype observed. We clarified this in the revised manuscript.

Rebuttal Figure 2: *KMT2C* expression in matched primary TNBC and metastases. Data was extracted from the same cohort used in Figure 1a.

6. Figure 4d shows *MMP3* is higher in TNBC with *KMT2C* mutations - what about in mets versus primary?

Thank you for raising this important point. In the same patient cohort as used in Fig. 1a, we compared *MMP3* expression in primary tumors (PT) vs metastases (mets) and found significantly lower *MMP3* levels in the metastases (**rebuttal Figure 3a**). However, this observation is

complicated by the fact that normal breast tissues have much higher *MMP3* expression than other organs (**rebuttal Figure 3b**). Thus, any contamination with normal breast tissue in the primary breast tumor samples would give the false impression that *MMP3* is decreased in metastases compared to primary tumors. Unfortunately, we do not have access to pure tumor epithelial cells from matched primary tumors and metastases from TNBC patients to conclusively answer this interesting question.

Rebuttal Figure 3: **a**, *MMP3* expression in matched primary tumors and metastases (data extracted from Siegel, M. B. et al, *J Clin Invest*, 2018). **b**, normalized TPM values of *MMP3* among different tissues, GTEx dataset.

7. Figure 4F.G - Were mets seen in any other locations?

We also analyzed bones of the mice used in this experiment. Similar to the data in Extended Data Fig 1p and q we found bone metastases in mice injected with *Kmt2c* or *Kmt2d_KO* cells. We did not see any difference in ratios of mice developing bone metastases upon *Mmp3* or *Kdm6a* knockdown in *Kmt2c* or *Kmt2d_KO* cells (**Extended Data Figure 5d**).

8. Why was only 2 days of GSK-J4 treatment used?

The rationale to treat only for two days is because two days of treatment was already sufficient to induce changes in histone modification (**Extended Data Figure 5f**) and *Mmp3* expression (**Fig. 4k**). Thus, to minimize indirect effects of the drug, we chose this early timepoint to focus on the

direct outcome of KDM6A inhibition. However, we now performed another *in vivo* experiment with longer *in vivo* treatment (12 days) essentially with the same results. See our response above to Reviewer 2's question 4.

9. what is the likelihood/path for translation?

We revised the following paragraph in the Discussion to elaborate on potential clinical translation in more detail:

“However, our finding that genetic or pharmacological inhibition of KDM6A reduces Mmp3 and prevents brain metastasis provides an alternative approach for therapeutic targeting of MMP3 in KMT2C or KMT2D-mutant tumors. As KDM6A inhibition seems to be well tolerated in mice, it could potentially be used in TNBC patients with KMT2C or KMT2D mutant tumors for the prevention of distant metastases in combination with currently used therapies. Alternative strategies to consider are combinations including P300 and BET bromodomain inhibitors since we observed a gain in in Kmt2c/Kmt2d KO cells and BET inhibitors have been shown to be more effective in Kmt2c KO cancer cells.”

10. Figure 2o - the images are very hard to see and thus not convincing

We added insets with higher magnification showing membranous staining of CD8 in the sections (**new Fig 2o**).

11. Can statistical analyses be performed on the immune changes assessed in Fig 2

We added p-values to the violin plots and revised the interpretation of the results in the manuscript text.

The abstract and MS are well written, but it could be improved if you could explain how this would be translated.

We revised the abstract following the reviewer's suggestion.

Decision Letter, first revision:

Our ref: NCB-A51987A

17th April 2024

Dear Dr. Polyak,

Thank you for submitting your revised manuscript "Loss of Kmt2c or Kmt2d drives brain metastasis via KDM6A-dependent upregulation of MMP3" (NCB-A51987A). It has now been seen by the original referees and their comments are below. The reviewers find that the paper has improved in revision, and therefore we'll be happy in principle to publish it in Nature Cell Biology, pending minor revisions to satisfy the referees' final requests and to comply with our editorial and formatting guidelines.

Thank you again for your interest in Nature Cell Biology Please do not hesitate to contact me if you have any questions.

Sincerely,

Zhe Wang, PhD
Senior Editor
Nature Cell Biology

Tel: +44 (0) 207 843 4924
email: zhe.wang@nature.com

Reviewer #2 (Remarks to the Author):

The authors have carefully responded to all of my comments with additional experiments and clarifications, and improved the manuscript with more mechanistic details. I do not have any further questions.

Reviewer #3 (Remarks to the Author):

Whilst most of my concerns were addressed I still feel

1. that they should have used MFP injections rather than experimental metastasis assays for all

experiments and maybe looked at using alternate models if met levels were not high enough
 2. they should have used pcr rather than FACS for the mcherry assessment. They have shown 10 correlations, but this is the first time FACS had been used and is not sufficient
 3. Still not clear in the text that their findings are brain met or met in general specific. I would argue if not brain specific they should use multiple models to show that multi-organ metastasis is impacted.
 4. MMP3 in primary tumour, mets and normal. Are there any additional existing databases/cohorts you can assess. Are there cohorts where the % of normal breast tissue contamination in primary tumors have been determined.

Author Rebuttal, first revision:

We thank the Editors and Reviewers for their critical reading of our revised manuscript and positive comments. We revised the manuscript following their suggestions. Below are our point-by-point responses (in blue) to each of the reviewers' specific points.

Reviewer #2:

Remarks to the Author:

The authors have carefully responded to all of my comments with additional experiments and clarifications, and improved the manuscript with more mechanistic details. I do not have any further questions.

We appreciate the reviewer's positive comments on our manuscript.

Reviewer #3:

Remarks to the Author:

Whilst most of my concerns were addressed I still feel

1. that they should have used MFP injections rather than experimental metastasis assays for all experiments and maybe looked at using alternate models if met levels were not high enough

Thank you for raising this point. We used MFP injection for the first experiments (Figure 1). But, due to the low frequency of metastases, we used intracardiac injection to be able to see

statistically significant quantitative differences between wild type and knock out cells. Intracardiac injection is a widely accepted method for quantifying differences in metastatic ability as reported in many other papers, for example Malladi, Srinivas et al. "Metastatic Latency and Immune Evasion through Autocrine Inhibition of WNT." Cell vol. 165,1 (2016) or Bos, Paula D et al. "Genes that mediate breast cancer metastasis to the brain." Nature vol. 459,7249 (2009). We believe that establishing additional models and repeating all experiments is beyond the scope of the current manuscript. In addition, almost all murine mammary tumors have very high spontaneous metastatic rate, thus, most would not be suitable for our studies. We tested multiple commonly used murine mammary tumor cell lines and 168FARN and 67NR cells are some of the few that are consistently form tumors but are not inherently metastatic.

2. they should have used pcr rather than FACS for the mcherry assessment. They have shown 10 correlations, but this is the first time FACS had been used and is not sufficient

We appreciate the reviewer's concern about the reliability of FACS for quantifying metastatic cells. However, as requested by the reviewer during the first revision, we provided data showing that quantification by FACS and qPCR correlate very well. Furthermore, FACS is a widely accepted method for the quantification of metastases formed by tumor cells expressing a fluorescent marker. It has been reported in many other papers, for example Ganesh, Karuna et al. "L1CAM defines the regenerative origin of metastasis-initiating cells in colorectal cancer." Nature cancer vol. 1,1 (2020).

3. Still not clear in the text that their findings are brain met or met in general specific. I would argue if not brain specific they should use multiple models to show that multi-organ metastasis is impacted.

Thank you for this suggestion. We clarified in the revised manuscript (last sentence of first Results section and also abstract), that the metastatic phenotype of the KO cells is not brain specific. We also see a difference in bone metastases, and there are micro-metastases in other organs as well (Figure 1f and Extended data figure 1p,q). We have shown this in two experimental models of TNBC. We believe that repeating all experiments in multiple additional models is beyond the scope of the current manuscript.

4. MMP3 in primary tumour, mets and normal. Are there any additional existing databases/cohorts you can assess. Are there cohorts where the % of normal breast tissue contamination in primary tumors have been determined.

We appreciate the reviewer's interest in this point. We have tried to analyze additional cohorts but did not identify any differences in MMP3 expression between primary and metastatic tumors. As we described before, MMP3 is highly and rather specifically expressed in the normal breast. Thus, primary breast tumors may have higher MMP3 expression due to contaminating with normal breast tissue making it impossible to see a difference between primary tumors and metastases based on bulk RNA-seq data. Importantly, we do not argue that sustained MMP3 expression is needed for the growth of established metastases, rather our data implies that it is important for metastatic seeding. We included the following sentence to our discussion to clarify this point.

'An unanswered question in our study is if Mmp3 only plays a role in metastatic seeding or if its expression is also sustained in established metastases.'

Final Decision Letter:

Dear Dr Polyak,

I am pleased to inform you that your manuscript, "Loss of Kmt2c or Kmt2d drives brain metastasis via KDM6A-dependent upregulation of MMP3", has now been accepted for publication in Nature Cell Biology.

Please note that *Nature Cell Biology* is a Transformative Journal (TJ). Authors may publish their research with us through the traditional subscription access route or make their paper immediately open access through payment of an article-processing charge (APC). Authors will not be required to make a final decision about access to their article until it has been accepted. Find out more about Transformative Journals

If you have not already done so, we strongly recommend that you upload the step-by-step protocols used in this manuscript to protocols.io (<https://protocols.io>), an open online resource that allows researchers to share their detailed experimental know-how. All uploaded protocols are made freely available and are assigned DOIs for ease of citation. Protocols and Nature Portfolio journal papers in which they are used can be linked to one another, and this link is clearly and prominently visible in the online versions of both. Authors who performed the specific experiments can act as primary authors for the Protocol as they will be best placed to share the methodology details, but the Corresponding

Author of the present research paper should be included as one of the authors. By uploading your Protocols onto protocols.io, you are enabling researchers to more readily reproduce or adapt the methodology you use, as well as increasing the visibility of your protocols and papers. You can also establish a dedicated workspace to collect your lab Protocols. Further information can be found at <https://www.protocols.io/help/publish-articles>.

With kind regards,

Zhe Wang, PhD
Senior Editor
Nature Cell Biology

Tel: +44 (0) 207 843 4924
email: zhe.wang@nature.com
